# MSF-DETR: A small target detection algorithm for sonar images based on spatial-frequency domain collaborative feature fusion

**Heng Zhao**[iD]**, Shuping Han, Jiaying Geng\*, Yubo Han, Shuyang Jia, Ke Li**

Naval Submarine Academy, Qingdao, China

\* zhao107932218@126.com

## Abstract

Side-scan sonar imaging is essential for underwater target detection in marine exploration and engineering applications, yet small target detection faces significant challenges including limited frequency domain feature utilization, insufficient multi-scale feature fusion, and high computational complexity. This study develops Multi-Scale Spatial-Frequency Collaborative Detection Transformer (MSF-DETR), a novel end-to-end automatic detection algorithm specifically designed for small targets in side-scan sonar images. The method integrates three core innovations: a Multi-domain Adaptive Spatial-frequency Network (MASNet) backbone employing Cascaded dual-domain Mamba-enhanced Spatial-frequency Synergistic Convolution that simultaneously captures spatial geometric and frequency domain texture features; a Hierarchical Multi-scale Adaptive Feature Pyramid Network implementing intelligent weight allocation across different scales; and an Efficient Sparse Attention Transformer Encoder utilizing Window-based Adaptive Sparse Self-Attention mechanism that reduces computational complexity from quadratic to linear. Experimental validation was conducted on the self-built SSST-3K(Side-Scan Sonar Target Detection 3K Dataset) dataset containing approximately 3000 high-quality sonar images and the public KLSG dataset. Results demonstrate that MSF-DETR achieves 78.5% mAP50 and 38.5% mAP50-95 on the SSST-3K dataset, representing improvements of 2.8% and 3.3% respectively compared to baseline RT-DETR, while reducing computational complexity by 12.0% and achieving 71.2 FPS inference speed. The proposed MSF-DETR provides an effective solution for small target detection in complex marine environments, significantly advancing underwater sonar image processing technology.

## Introduction

The automatic detection technology for small targets in side-scan sonar images holds significant application value and strategic importance in marine exploration, underwater navigation, marine engineering, and marine biological monitoring.

**Data availability statement:** The dataset used in this study is publicly and freely available without restrictions from the GitHub repository: https://github.com/huoguanying/SeabedObjects-Ship-and-Airplane-dataset.

**Funding:** The author(s) received no specific funding for this work.

**Competing interests:** The authors have declared that no competing interests exist.

As an acoustic sensor capable of overcoming the limitations of optical imaging in turbid and low-light underwater environments, side-scan sonar provides reliable technical means for underwater target detection [1]. With the growing demands for marine resource development, underwater infrastructure maintenance, and marine security monitoring, accurate and efficient small target detection algorithms have become key technologies for advancing intelligent marine equipment development. Zhou et al. [2] proposed an automatic detection method for small targets in side-scan sonar images that demonstrated excellent detection performance in complex marine environments, proving the important value of this technology in practical engineering applications. Therefore, developing high-precision small target detection algorithms tailored to the characteristics of side-scan sonar images holds significant theoretical meaning and practical value for enhancing the intelligence level of underwater autonomous operation systems, ensuring marine engineering safety, and promoting marine scientific research.

Traditional side-scan sonar image target detection methods primarily rely on manually designed feature extraction and pattern recognition techniques, achieving target identification by analyzing statistical characteristics, texture features, and geometric shapes of sonar images. Abu et al. [3] proposed a statistical feature extraction method based on weighted likelihood ratios, combined with support vector machine classifiers to achieve sonar image target recognition, which achieved good detection results in specific scenarios. Chen et al. [4] developed an underwater target detection method based on spectral residual and three-frame algorithms, enhancing target detection robustness by combining frequency domain analysis with temporal information. However, these traditional methods generally suffer from limited feature expression capabilities, poor adaptability to complex marine environments, and requirements for extensive manual parameter tuning, making them difficult to meet modern marine engineering demands for detection accuracy and real-time performance.

In recent years, the rapid development of deep learning technology has brought new breakthroughs to side-scan sonar image target detection. Wang et al. [5] proposed a weak small target detection method for side-scan sonar images based on multi-branch shuttle neural networks, significantly improving small target detection accuracy through multi-scale feature fusion. Zhang et al. [6] developed a side-scan sonar image target detection model based on improved YOLOv5, introducing transfer learning techniques to overcome the problem of scarce sonar image data. Li et al. [7] proposed an advanced deep learning framework based on YOLOv7 architecture, achieving high-precision target detection for multibeam side-scan sonar through optimization of data preprocessing, feature fusion, and loss functions. Fan et al. [8] designed a side-scan sonar image target detection and segmentation algorithm based on improved Mask R-CNN, effectively handling target boundary problems in complex underwater environments.

Recently, deep learning-based object detection methods have achieved remarkable progress in computer vision, demonstrating mature technical systems and powerful real-time processing capabilities. Modern object detection frameworks such as YOLO series and DETR series excel in handling complex scenes, multi-scale targets, and dense target distributions, providing strong technical support for small

target detection in side-scan sonar images. Therefore, this study adopts object detection methods as the technical approach, aiming to fully utilize their advantages in accuracy and efficiency to solve key technical challenges in small target detection for side-scan sonar images.

In the field of sonar image processing based on object detection, researchers have conducted extensive exploratory work. Li et al. [9] proposed CCW-YOLOv5, a side-scan sonar target detection method based on coordinate convolution and improved bounding box loss, significantly enhancing small target detection capability by introducing position information-rich feature extraction. Yu et al. [10] developed a side-scan sonar image target detection model based on Transformer-YOLOv5, improving model adaptability to complex seafloor terrain through self-attention mechanisms. With the introduction of DETR (Detection Transformer) architecture, researchers began exploring applications of end-to-end detection methods in sonar image processing. Wang et al. [11] proposed US-DETR, an underwater sonar image target detection method based on improved RT-DETR, significantly improving detection performance through enhanced feature interaction modules and non-local attention feature fusion mechanisms. Chen et al. [12] developed NAS-DETR, a method based on zero-shot neural architecture search, achieving excellent performance in sonar target detection tasks. Recent research work also includes ProNet proposed by Wang et al. [13], a network based on progressive sensitivity capture, and multiple improvement works based on RT-DETR architecture optimization, providing important technical accumulation and theoretical foundation for RT-DETR [14] applications in sonar image processing.

Although object detection-based methods and advanced algorithms like RT-DETR perform excellently in general object detection tasks, they still face numerous technical challenges in small target side-scan sonar image automatic detection applications. First, side-scan sonar images have unique imaging mechanisms and signal characteristics, making traditional spatial domain feature extraction methods unable to fully utilize frequency domain information of sonar signals, resulting in insufficient and inaccurate feature representation of small targets. Second, small targets in sonar images often exhibit characteristics such as large scale variations, blurred boundaries, and susceptibility to seafloor reverberation interference, while existing feature fusion networks lack adaptive integration mechanisms for multi-scale small target features, making it difficult to achieve stable detection performance in complex marine environments. Additionally, existing public sonar datasets are limited in scale and variable in quality, lacking dedicated high-quality datasets for side-scan sonar small target detection, which severely restricts algorithm development and performance evaluation. Finally, traditional Transformer encoders suffer from high computational complexity and insufficient spatial structure modeling capabilities when processing high-resolution sonar images, affecting model real-time performance and effective capture of small target spatial features. These technical bottlenecks severely restrict the application effectiveness of existing algorithms in practical marine engineering.

To address the aforementioned technical challenges, this paper proposes MSF-DETR, a small target side-scan sonar image automatic detection algorithm that effectively solves the technical limitations of traditional methods in complex marine environments through constructing a dedicated dataset and three core innovative modules working collaboratively. The main contributions of this paper are as follows:

(1) Construction of a high-quality side-scan sonar small target detection dataset named SSST-3K (Side-Scan Sonar Target Detection 3K Dataset). Through data collection by surface vessels towing side-scan sonar in real marine environments, a dedicated dataset containing approximately 3000 high-quality sonar images was constructed. The dataset employs high-low frequency dual-mode operation to obtain multi-frequency target features, covering 3 typical small target types, and provides important benchmark data support for side-scan sonar small target detection algorithm development and evaluation through professional annotation and scientific partitioning.

(2) Proposal of Multi-domain Adaptive Spatial-frequency Network (MASNet) backbone network that employs Cascaded dual-domain Mamba-enhanced Spatial-frequency Synergistic Convolution (CMSSC) feature extraction modules and Dual-domain Spatial-frequency Synergistic Convolution (DSSC) collaborative convolution mechanisms to achieve simultaneous capture and fusion of spatial geometric features and frequency domain texture information in sonar images, significantly enhancing multi-modal feature representation capabilities for small targets.

(3) Design of Hierarchical Multi-scale Adaptive Feature Pyramid Network (HMAFPN), a multi-scale feature fusion network that achieves intelligent weight allocation and optimal combination of different scale features through Multi-input Adaptive Fusion Module (MAFM), breaking through the limitations of traditional FPN unidirectional information flow and effectively improving the expression richness and fusion effects of small target features.

(4) Proposal of Efficient Sparse Attention Transformer Encoder (ESATE) that employs Window-based Adaptive Sparse Self-Attention (WASSA) mechanism to reduce computational complexity from quadratic to linear scale, while using Spatial-Enhanced Feed-Forward Network (SEFFN) to replace traditional feed-forward networks, significantly improving spatial structure modeling capabilities and computational efficiency for sonar images, achieving optimal balance between accuracy and speed.

## Related work

### Introduction to self-built dataset SSST-3K

This study employs the self-built SSST-3K dataset to validate the effectiveness of the proposed algorithm. The dataset was obtained through forward-looking sonar surveys of small targets conducted by surface vessels towing equipment at sea, possessing high practicality and representativeness. To improve survey efficiency and data quality, the forward-looking sonar employs high-low frequency dual-mode operation, capable of displaying both high-frequency and low-frequency sonar images separately, thus obtaining target feature information under different frequency bands. Meanwhile, to increase the receptive field and obtain more complete target information, this study captures full-screen waterfall images as detection images, ensuring complete target contours and surrounding environmental information can be captured. Dataset examples are shown in Fig 1.

After standardized processing, the SSST-3K dataset contains approximately 3000 sonar images with targets, covering three different types of small targets: cone (MT), cylinder (C), and sphere (M). All targets were precisely marked using professional labelImg annotation software, constructing a complete sonar image dataset. To ensure good generalization capability of the detection model, the dataset was divided into training, validation, and test sets according to a 7:1:2 ratio, with the training set containing 2182 images, validation set containing 312 images, and test set containing 626 images.

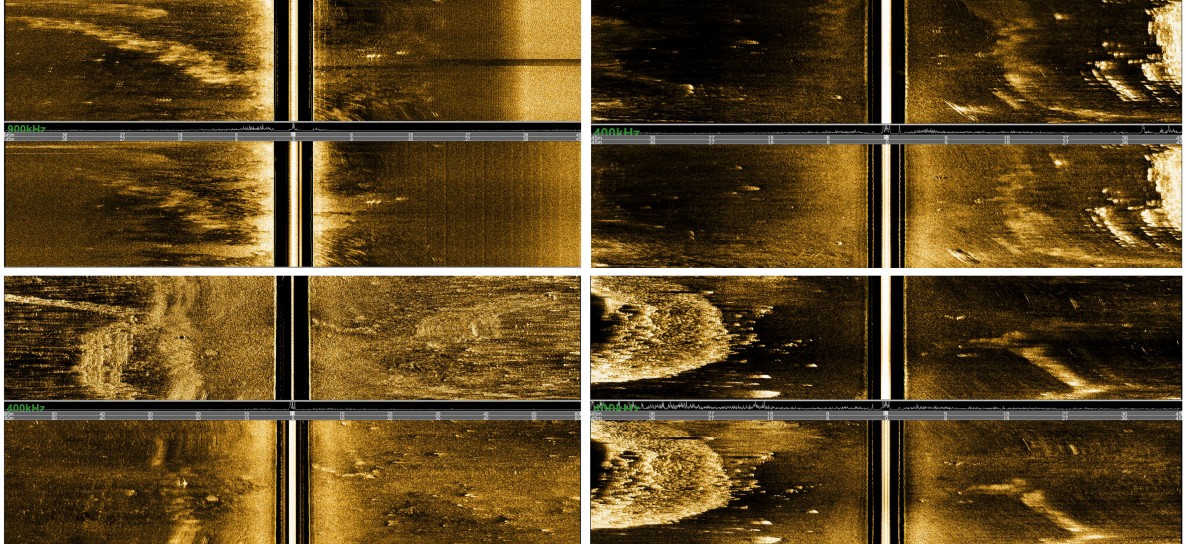

**Fig 1**. Examples of SSST-3K dataset containing cone, cylinder, and sphere targets.

To further improve detection result reliability and model discrimination capability for negative samples, over 600 background images without targets were added to the test set in a 1:1 ratio, making the actual test set scale reach 1252 images, thus constructing a balanced and challenging evaluation benchmark.

### RT-DETR baseline framework

RT-DETR (Real-Time Detection Transformer) is the first real-time end-to-end object detection algorithm that successfully addresses the problems of high computational cost and slow inference speed in traditional DETR series algorithms while maintaining high detection accuracy and achieving real-time performance. The core innovation of RT-DETR lies in designing an efficient hybrid encoder architecture that significantly reduces computational complexity by decoupling intra-scale interaction and cross-scale fusion operations. The algorithm employs convolutional neural networks as the backbone network for feature extraction, extracting multi-scale features from the last three stages $\{S3, S4, S5\}$ of the backbone network as encoder inputs. This design ensures feature richness while controlling computational overhead.

The encoder part of RT-DETR contains two key modules: Attention-based Intra-scale Feature Interaction (AIFI) module and CNN-based Cross-scale Feature Fusion (CCFF) module. The AIFI module is specifically responsible for processing high-level semantic features from the S5 layer, performing intra-scale feature interaction through single-scale Transformer encoders to effectively capture semantic associations in high-level features. The CCFF module is responsible for fusion between different scale features, achieving cross-scale information transfer through convolution operations. The decoder part adopts standard Transformer decoder structure equipped with auxiliary prediction heads, providing high-quality initial target queries for the decoder through IoU-aware query selection mechanisms, and finally generating final detection results through iterative optimization. This end-to-end design eliminates non-maximum suppression (NMS) post-processing steps, not only simplifying the detection pipeline but also improving inference speed and detection accuracy.

### Related work on feature fusion networks

Feature Pyramid Network (FPN), as a classic architecture for multi-scale feature fusion, plays an important role in object detection. Lin et al. initially proposed FPN through top-down pathways and lateral connections to construct multi-scale feature representations [15], but using only unidirectional information flow has certain limitations. To address this problem, Liu et al. proposed Path Aggregation Network (PANet), enhancing information transfer from low-level to high-level features by adding bottom-up pathways [16]. Subsequently, Ghiasi et al. proposed NAS-FPN based on neural architecture search, discovering better feature fusion topological structures through automatic search [17]. Tan et al. proposed Bidirectional Feature Pyramid Network (BiFPN) in EfficientDet, achieving better balance between efficiency and accuracy through weighted bidirectional feature fusion and removal of redundant connections [18]. In recent years, researchers have further explored applications of attention mechanisms in feature fusion. Dang et al. proposed Hierarchical Attention Feature Pyramid Network (HA-FPN), enhancing feature expression capabilities by introducing Transformer and channel attention modules [19]. Chen et al. proposed Improved Feature Pyramid Network (ImFPN) that further optimizes feature fusion effects through similarity fusion modules and attention mechanisms [20]. These works provide important theoretical foundation and technical reference for the hierarchical multi-scale adaptive feature pyramid network proposed in this paper.

### Methods

This paper proposes MSF-DETR (Spatial-Frequency Domain collaborative DETR) algorithm targeting the special challenges of small target detection tasks in side-scan sonar images, constructing an end-to-end detection framework. The framework includes three core innovations: first is the MASNet backbone network that enhances small target feature representation through spatial-frequency dual-domain collaborative processing mechanisms; second is the HMAFPN feature fusion network that optimizes multi-scale feature integration using dense cross-layer connections and adaptive weight

fusion strategies; finally is the ESATE encoder that improves feature encoding efficiency through sparse attention mechanisms and spatially-aware feed-forward networks. The overall framework structure is shown in Fig 2. The three innovative modules work collaboratively to effectively solve problems of insufficient detection accuracy and low computational efficiency for small targets in complex marine environments using traditional methods.

## Multi-domain Adaptive Spatial-frequency Network (MASNet) backbone design

Traditional ResNet [21] backbone networks face technical bottlenecks in side-scan sonar image small target detection. First, ResNet's fixed convolution kernel design cannot effectively handle frequency domain feature distribution differences of targets in sonar images. Under high-low frequency hybrid working modes, single spatial domain feature extraction strategies have difficulty capturing complete spectral information of sonar signals, leading to insufficient frequency domain feature representation of small targets. Second, ResNet lacks adaptive gating mechanisms for sonar image noise characteristics. In complex seafloor reverberation and multipath interference environments, networks cannot dynamically adjust weight allocation of feature channels, making them susceptible to strong background noise interference. Therefore, this paper proposes MASNet backbone network based on CMSSC feature extraction modules. This network introduces gated spatial-frequency dual-domain feature fusion mechanisms and adaptive channel selection strategies, enabling simultaneous feature extraction and fusion in both spatial and frequency domains, effectively solving frequency domain information loss problems in traditional networks for sonar image processing. Through collaborative action of DSSC modules, detection accuracy and robustness for complex small targets in marine environments are significantly improved.

The overall structure of the CMSSC module is shown in Fig 3, employing hierarchical feature extraction and gated fusion design concepts. The module first performs channel expansion transformation on input features $X \in \mathbb{R}^{B \times C_1 \times H \times W}$, expanding input channels from $C_1$ to $2C$ through $1 \times 1$ convolution. The expanded features are equally divided into two

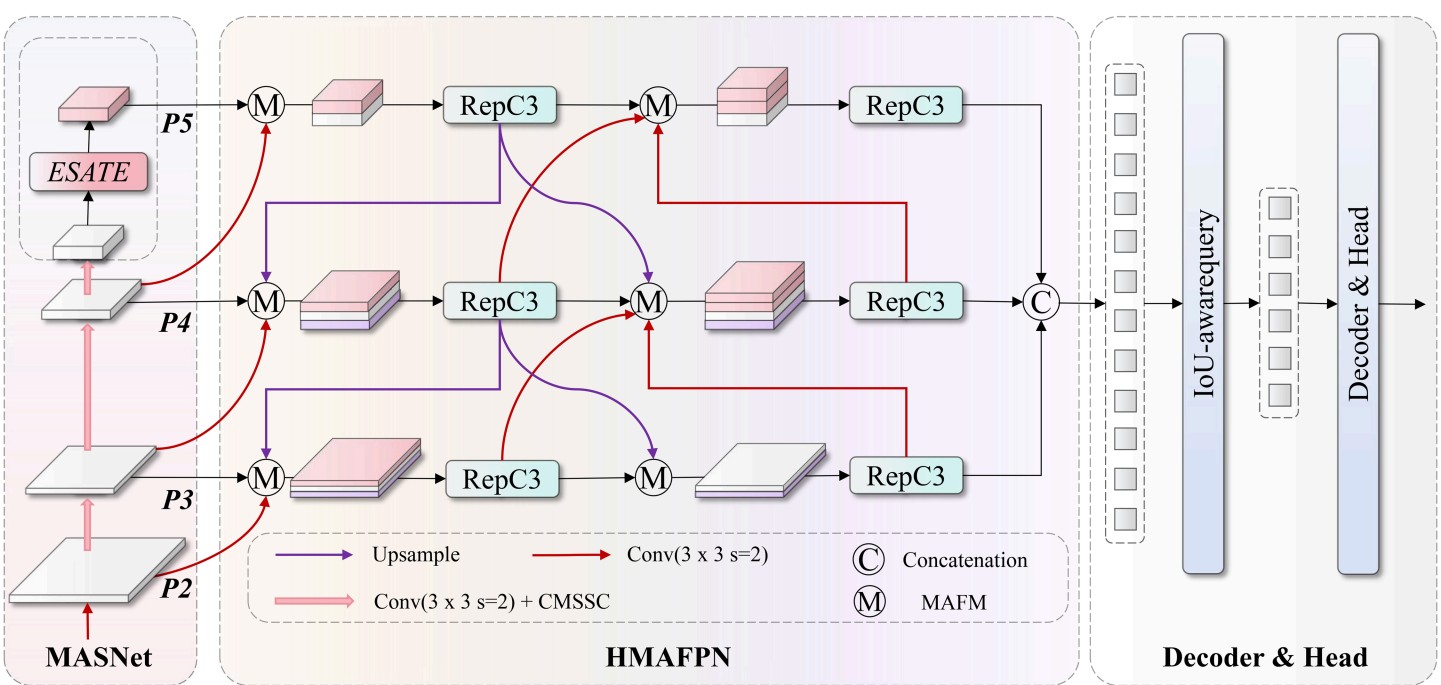

**Fig 2. Overall framework architecture of MSF-DETR.** The framework integrates MASNet backbone, HMAFPN feature fusion network, and ESATE encoder to achieve end-to-end small target detection in side-scan sonar images.

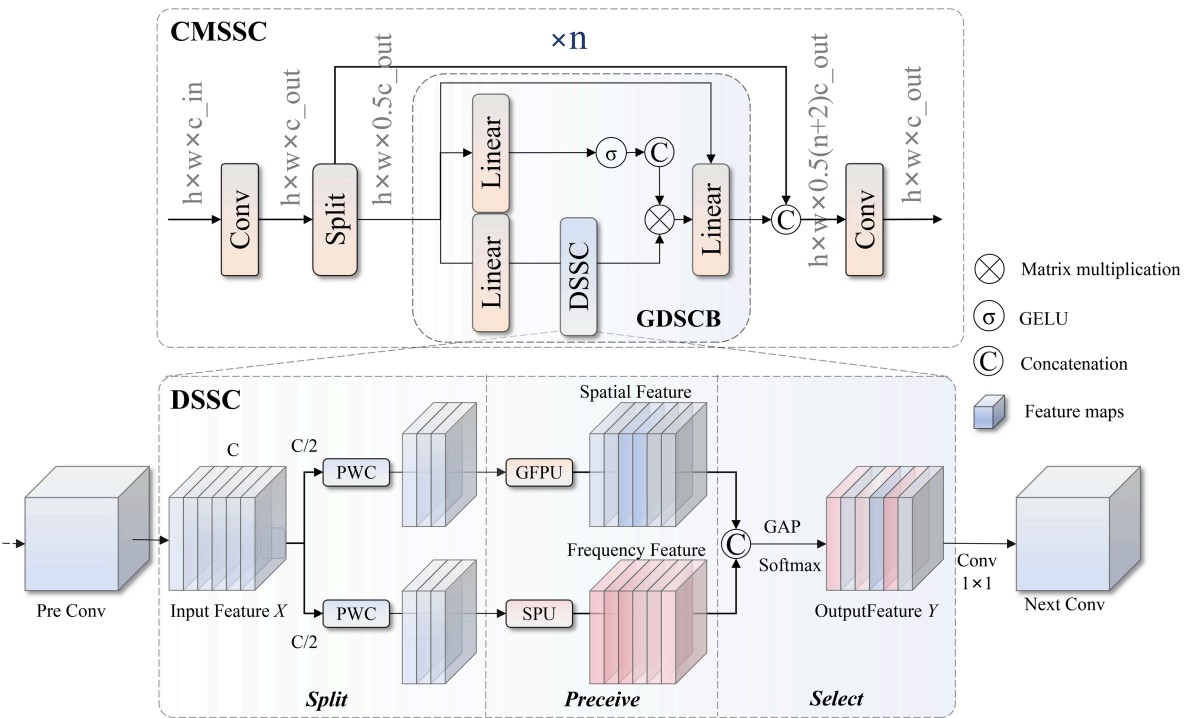

**Fig 3**. Architecture diagram of the CMSSC module.

sub-features $X_{s1}$ and $X_{s2}$ in the channel dimension, where $X_{s1}$ directly participates in final feature connection operations, while $X_{s2}$ serves as recursive input for subsequent $n$ Gated Dual-domain Synergistic Convolution Block (GDSCB) modules. The module progressively refines feature representations through cascading multiple gated DSSC blocks. Its overall mathematical expression can be described as:

$$Y = \phi_{conv}^{2C}\left(X_{s1}, X_{s2} \cup \{G_i(F_{i-1})|i = 1, 2, \ldots, n\}\right) \tag{1}$$

where $\phi_{conv}^{2C}(\cdot)$ represents output convolution transformation, $C(\cdot)$ represents channel concatenation operation, $G_i(\cdot)$ represents the $i$-th GDSCB module, $F_0 = X_{s2}$ is the initial recursive input, and $n$ is the number of module repetitions.

The GDSCB module performs layer normalization on input features to improve training stability, mapping features to high-dimensional representation space through fully connected layers with expansion ratio $\alpha = 8/3$. The expanded features are decomposed into three functionally different sub-feature branches according to preset ratios $h$, $h - c_{conv}$, $c_{conv}$ in the channel dimension: gating signal branch $G$, identity mapping branch $I$, and DSSC convolution branch $C$. After GELU activation function processing, the gating signal performs gated fusion with the identity mapping branch and convolution branch processed by DSSC. The mathematical expression for this process is:

$$F_{out} = F_{in} + D\left(\phi_{fc2}\left(\sigma_{GELU}(G) \odot C(I, S_{DSSC}(C))\right)\right) \tag{2}$$

where $G, I, C = S(\phi_{fc1}(N_{LN}(F_{in})))$, $S(\cdot)$ represents channel splitting operation, $N_{LN}(\cdot)$ represents layer normalization, $\phi_{fc1}$ and $\phi_{fc2}$ represent input and output fully connected layers respectively, $\sigma_{GELU}(\cdot)$ is the GELU activation function, $S_{DSSC}(\cdot)$

represents DSSC spatial-frequency collaborative convolution operation, $D(\cdot)$ represents DropPath random depth regularization, and $\odot$ represents Hadamard element-wise multiplication.

The DSSC module adopts dual-branch parallel processing architecture: Spatial Processing Unit (SPU) is responsible for extracting spatial domain geometric features, while Gabor-enhanced Frequency Processing Unit (GFPU) specifically handles frequency domain texture and edge information. Output features from both branches are integrated through adaptive weight fusion mechanisms, finally achieving dynamic feature selection through soft attention mechanisms. The overall mathematical expression for DSSC can be described as:

$$F_{spa} = U_{SPU}(\phi_{PWC}^0(X))$$
$$F_{freq} = U_{GFPU}(\phi_{PWC}^1(X))$$
$$F_{DSSC} = \phi_{PWC}^o(\sigma_{softmax}(P_{avg}[F_{spa}, F_{freq}]))$$

(3)

where $\phi_{PWC}^0$, $\phi_{PWC}^1$ and $\phi_{PWC}^o$ represent input and output pointwise convolution layers respectively, $U_{SPU}(\cdot)$ and $U_{GFPU}(\cdot)$ represent spatial processing unit and Gabor frequency processing unit respectively, $[\cdot]$ represents concatenation operation, $P_{avg}(\cdot)$ represents adaptive average pooling operation, and $\sigma_{softmax}(\cdot)$ is the softmax normalization function. This design enables simultaneous capture of spatial geometric information and spectral texture features in sonar images through parallel dual-domain processing and adaptive fusion strategies.

The SPU module is specifically responsible for extracting spatial domain features of sonar images, employing branch cascading and progressive fusion processing strategies. The module first equally divides input features into two sub-features $X_1$ and $X_2$ in the channel dimension, then employs depth-wise separable convolution kernels of different sizes for feature extraction. $X_1$ extracts local spatial features through $3 \times 3$ depth convolution, while $X_2$ first performs residual connection with $X_1$ output, then extracts larger receptive field spatial features through $5 \times 5$ depth convolution. This cascading design effectively captures spatial geometric information at different scales. The mathematical expression for SPU is:

$$X_1' = \phi_{dwconv}^{3 \times 3}(X_1)$$
$$X_2' = \phi_{dwconv}^{5 \times 5}(X_2 + X_1')$$
$$F_{out} = \phi_{conv}^{1 \times 1}([X_1', X_2']) + X_{res}$$

(4)

where $\phi_{dwconv}^{3 \times 3}(\cdot)$ and $\phi_{dwconv}^{5 \times 5}(\cdot)$ represent $3 \times 3$ and $5 \times 5$ depth-wise separable convolutions respectively, $\phi_{conv}^{1 \times 1}(\cdot)$ represents $1 \times 1$ pointwise convolution for feature fusion, and $X_{res}$ represents the residual connection term. This design effectively extracts geometric shape and spatial structure features of targets in sonar images through cascaded processing of multi-scale spatial convolutions.

The GFPU module is an innovative component specifically designed for processing frequency domain features of sonar images, based on multi-directional and multi-scale characteristics of Gabor filters to extract texture and edge information from images. The module first equally divides input features into four sub-channel groups, with each sub-channel group processed through independent GaborSingle filters to capture frequency domain features at different orientation angles $\theta \in \{0°, 45°, 90°, 135°\}$ and different scales $s \in \{1, 2, 3, 4\}$. Output features from the four sub-channels are concatenated and then integrated through fully connected layers for feature integration and dimensionality adjustment. The mathematical expression for GFPU is:

$$X_1, X_2, X_3, X_4 = \text{Split}(X, \dim = 1)$$
$$F_{gabor} = C([G_{single}(X_i) | i = 1, 2, 3, 4])$$
$$F_{out} = \phi_{fc}(F_{gabor}) + X_{res}$$

(5)

where $G_{single}(\cdot)$ represents single Gabor filter operation based on multi-angle, multi-scale Gabor kernel convolution calculations: $G_{single}(X) = \sum_{\theta,s} w_{\theta,s} * X * G_{\theta,s}(x, y)$, where $G_{\theta,s}(x, y)$ is the parameterized Gabor filter kernel, $w_{\theta,s}$ are learnable weight parameters, and * represents convolution operation. $\phi_{fc}(\cdot)$ represents fully connected layers for feature integration, and $X_{res}$ is the residual connection term. This design effectively extracts frequency domain texture features and edge contour information of targets in sonar images, particularly suitable for processing underwater small targets with complex scattering characteristics.

The proposed MASN backbone network effectively solves the fundamental problem of insufficient frequency domain information utilization when traditional ResNet processes sonar images through introducing gated dual-domain feature fusion strategies and adaptive channel selection mechanisms, significantly improving multi-modal feature representation capabilities for small targets in complex marine environments. Particularly through the spatial-frequency parallel processing design of DSSC modules, the network can simultaneously capture geometric spatial features and spectral texture information of sonar images, effectively utilizing physical characteristics of sonar signals for feature enhancement. Through gating mechanisms and soft attention strategies, different domain feature weight contributions are dynamically adjusted, greatly improving accuracy and robustness of small target detection.

## Hierarchical Multi-scale Adaptive Feature Pyramid Network (HMAFPN) design

Traditional Cross-scale Cascade Feature Merging (CCFM) feature fusion networks employ simple upsampling-concatenation-convolution linear fusion strategies, exhibiting significant limitations when processing multi-scale features of side-scan sonar images. First, CCFM adopts direct feature concatenation for multi-scale information fusion, lacking adaptive evaluation mechanisms for the importance of different scale features, causing high-value small target features to be easily masked by large-scale background features. Second, traditional top-down unidirectional information flow design ignores the feedback enhancement effect of bottom-level detail features on high-level semantic features, failing to fully exploit complementary relationships between multi-layer features. Finally, CCFM lacks dense connection mechanisms across hierarchical levels, resulting in insufficient information interaction between distant feature layers, limiting network expression capabilities for complex sonar scenes. Addressing these problems, this paper proposes HMAFPN feature fusion network architecture that constructs adaptive multi-scale feature fusion mechanisms through introducing MAFM and dense cross-layer connection strategies, intelligently evaluating and integrating contribution weights of different hierarchical features, significantly enhancing feature representation capabilities for small targets in sonar images, effectively solving problems of insufficient feature information utilization in traditional fusion networks under complex marine environments, providing more powerful and robust feature representation foundations for side-scan sonar image small target detection. The structure of HMAFPN is shown in Fig 4.

HMAFPN adopts densely connected feature pyramid architecture, achieving efficient integration of multi-scale features through multiple feature fusion pathways and adaptive attention mechanisms. The network breaks through limitations of traditional FPN unidirectional information flow, constructing bidirectional multi-path feature propagation mechanisms that enable each feature layer to simultaneously receive enhancement from upper-level semantic information and lower-level detail information. The core innovation of the network lies in introducing MAFM modules as basic units for feature fusion. These modules can adaptively learn importance weights of different input features, achieving optimal feature combinations through attention mechanisms. The overall feature fusion process of HMAFPN can be described through hierarchically recursive mathematical expressions that reflect the network's multi-path information aggregation characteristics:

$$F_l^{out} = \mathcal{M}_{MAFM}(F_l^{in}, U(F_{l+1}^{out}), D(F_{l-1}^{out}), \mathcal{L}(F_{l-k}^{backbone})) \tag{6}$$

where $F_l^{out}$ represents the output feature of the $l$-th layer, $\mathcal{M}_{MAFM}(\cdot)$ represents the operation function of the multi-input adaptive fusion module, $U(\cdot)$ and $D(\cdot)$ represent upsampling and downsampling operations respectively, $\mathcal{L}(\cdot)$ represents

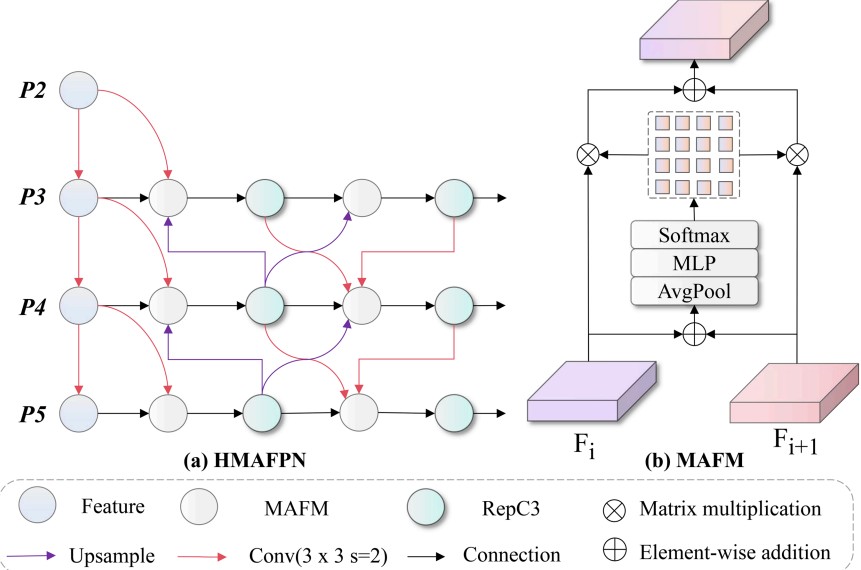

**Fig 4**. **Architecture of HMAFPN.** The network implements dense cross-layer connections and MAFM.

lateral connections from the backbone network, and $k$ represents the distance of cross-layer connections. This design enables each feature layer to receive information enhancement from multiple directions and multiple scales, significantly improving feature expression richness.

HMAFPN constructs a fully interconnected feature fusion network through building dense cross-layer connection networks, achieving comprehensive information interaction between all hierarchical levels of the feature pyramid. The network not only maintains traditional FPN top-down pathways but also adds bottom-up feedback pathways and cross-hierarchical direct connection pathways, forming a highly interconnected feature fusion network. This dense connection design can be mathematically described using graph theory, where each node represents a feature layer and edges represent information propagation paths:

$$G_{HMAFPN} = (V, \mathcal{E}), \quad \mathcal{E} = \mathcal{E}_{TD} \cup \mathcal{E}_{BU} \cup \mathcal{E}_{Skip} \cup \mathcal{E}_{Cross} \tag{7}$$

where $V = \{F_3, F_4, F_5\}$ represents the set of feature nodes, and $\mathcal{E}_{TD}, \mathcal{E}_{BU}, \mathcal{E}_{Skip}, \mathcal{E}_{Cross}$ represent top-down edges, bottom-up edges, skip connection edges, and cross-layer connection edges respectively. This graph structure design ensures that each feature layer in the network can receive information from all other hierarchical levels. Through adaptive fusion mechanisms of MAFM modules, the network can dynamically adjust information weights from different pathways according to specific input content, thereby achieving optimal feature representation effects.

The MAFM module can handle arbitrary numbers of input feature maps, achieving intelligent feature selection and fusion by learning importance weights for each input feature. MAFM first unifies dimensions of different input features through a series of $1 \times 1$ convolutions, ensuring all input features have the same channel dimensions for subsequent fusion operations. Then the unified feature maps are stacked in the channel dimension, forming a four-dimensional tensor where the second dimension represents different input feature sources. Next, the module extracts global context information for each feature map through global average pooling and generates corresponding attention weights through

multi-layer perceptrons. This process can be described as multi-input feature adaptive weighted combination:

$$F_{unified} = C_{stack}([\phi_{proj}^i(F_i^{in})|i = 1, 2, \dots, N]) \in \mathbb{R}^{B \times N \times C \times H \times W} \tag{8}$$

where $\phi_{proj}^i(\cdot)$ represents the projection function for the $i$-th input feature, which is a $1 \times 1$ convolution when input channel numbers differ from target dimension $C$, otherwise an identity mapping, $C_{stack}(\cdot)$ represents stacking operation in a new dimension, $N$ is the number of input features, and $B$, $C$, $H$, $W$ represent batch size, channel number, height, and width respectively. This unification processing ensures that features from different sources can be effectively fused in the same representation space.

The MAFM mechanism learns importance weights for each input feature through global information aggregation and nonlinear transformation. The module first sums the stacked features across the feature dimension to obtain initial fused features, then compresses spatial dimensions to $1 \times 1$ through global average pooling to extract global context information. Next, attention weights are generated through a multi-layer perceptron structure containing dimensionality reduction-activation-dimensionality expansion, and finally normalized using softmax function to ensure all weights sum to 1. The mathematical expression for this process is:

$$F_{global} = GAP\left(\sum_{i=1}^{N} F_{unified}[:, i, :, :, :]\right) \in \mathbb{R}^{B \times C \times 1 \times 1}$$

$$A_{weights} = \text{Softmax}(\phi_{MLP}(F_{global})) \in \mathbb{R}^{B \times N \times C \times 1 \times 1} \tag{9}$$

where $GAP(\cdot)$ represents global average pooling operation, $\phi_{MLP}(\cdot)$ represents multi-layer perceptron, typically containing $\phi_{conv1}(\cdot) \circ \sigma_{ReLU} \circ \phi_{conv2}(\cdot)$ structure, where the first convolution layer reduces channel numbers from $C$ to $\max(C/r, 4)$ (where $r$ is the reduction ratio), the second convolution layer expands channel numbers to $N \times C$, and the softmax function performs normalization in the feature dimension $N$. The final fused feature is obtained through weighted summation: $F_{out} = \sum_{i=1}^{N}(A_{weights}[:, i, :, :, :] \odot F_{unified}[:, i, :, :, :])$, where $\odot$ represents broadcast element-wise multiplication.

The proposed HMAFPN multi-scale feature fusion network significantly improves feature representation capabilities for small target detection in side-scan sonar images through innovative dense connection architecture and adaptive attention fusion mechanisms. The network breaks through limitations of traditional FPN unidirectional information flow by constructing bidirectional multi-path feature propagation networks and introducing MAFM adaptive fusion modules, achieving full utilization and intelligent integration of multi-scale feature information.

## Efficient Sparse Attention Transformer Encoder (ESATE) design

Traditional TransformerEncoderLayer faces significant performance bottlenecks when processing side-scan sonar image feature encoding. First, standard global self-attention mechanisms have $O(N^2)$ computational complexity, generating enormous computational overhead when processing high-resolution sonar images, severely affecting real-time detection efficiency. Second, traditional feed-forward networks (FFN) rely only on simple linear transformations and point activation functions, lacking effective modeling capabilities for spatial geometric structures, performing poorly when processing complex seafloor terrain and target shapes in sonar images. Finally, traditional encoders lack multi-scale feature adaptive fusion mechanisms, unable to fully utilize semantic information from different hierarchical levels, resulting in insufficient and inaccurate feature representation of small targets. Addressing these limitations, this paper proposes ESATE encoder that reduces computational complexity from quadratic to linear scale through introducing WASSA mechanism, while SEFFN module significantly enhances spatial structure modeling capabilities for sonar images through multi-scale spatial branches and adaptive feature fusion strategies, effectively solving efficiency and accuracy problems of traditional encoders in small target detection tasks. The structure of HMAFPN is shown in Fig 5.

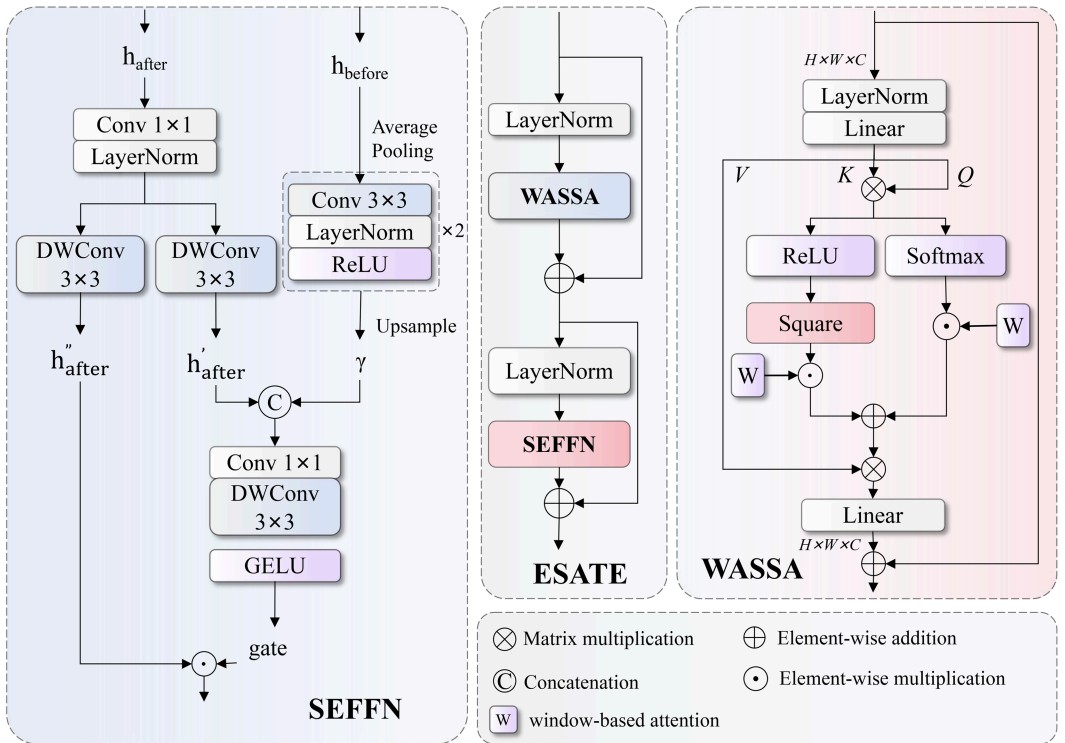

**Fig 5. Structure of ESATE.** The encoder incorporates WASSA mechanism and SEFFN.

ESATE encoder adopts dual-stage feature processing architecture, achieving efficient encoding of sonar image features through serial collaboration of adaptive sparse attention and spatial-enhanced feed-forward networks. The encoder first converts input feature maps from spatial format to sequence format for attention mechanism processing, then performs windowed sparse attention computation through WASSA module, effectively modeling local and global spatial dependencies. Subsequently, the encoder employs SEFFN module to replace traditional feed-forward networks, enhancing spatial awareness capabilities of features by introducing original spatial feature branches. The entire processing strictly follows Transformer design principles, including residual connections and layer normalization operations, ensuring training stability and effective gradient propagation. The mathematical expression of the encoder embodies its dual-stage processing core concept:

$$F_{out} = \mathcal{L}N_2(F_{attn} + D_2(\mathcal{F}_{SEFFN}(F_{attn}, F_{in}))) \tag{10}$$

where $F_{attn} = \mathcal{L}N_1(F_{in} + D_1(A_{WASSA}(F_{in})))$ represents intermediate features processed by adaptive sparse attention and first normalization, $A_{WASSA}(\cdot)$ represents windowed adaptive sparse self-attention function, $\mathcal{F}_{SEFFN}(\cdot, \cdot)$ represents dual-input processing function of spatial-enhanced feed-forward network, $\mathcal{L}N_1$ and $\mathcal{L}N_2$ represent two layer normalization operations respectively, and $D_1$ and $D_2$ represent corresponding random depth dropout operations.

The WASSA mechanism achieves balance between computational efficiency and modeling capability through windowed partitioning and cyclic shifting strategies. The mechanism first reshapes input feature maps to sequence format, then divides feature sequences into multiple non-overlapping window blocks according to predefined window sizes. To maintain information exchange between different windows, the mechanism introduces cyclic shifting operations, achieving

cross-window feature interaction by periodically moving window boundaries. This design reduces global attention computational complexity from $O(H^2 W^2)$ to $O(HW \cdot win^2)$, where $win$ represents window size. The mathematical expressions for window partitioning and cyclic shifting are:

$$X_{win} = W_{partition}(S_{roll}(F_{norm}, -s_{shift}, -s_{shift})) \in \mathbb{R}^{N_w \times win^2 \times C} \tag{11}$$

where $S_{roll}(\cdot, \cdot)$ represents cyclic shifting function, shifting $-s_{shift}$ pixels along height and width dimensions respectively, $W_{partition}(\cdot)$ represents window partitioning operation, dividing shifted feature maps into $N_w = \frac{HW}{win^2}$ windows, $s_{shift}$ is the shifting distance, typically set to $\frac{win}{2}$ to achieve optimal information exchange effects.

Within each window, WASSA employs sparsified attention computation strategy, retaining only the most important attention connections through top-k selection mechanism, further reducing computational overhead. This mechanism not only reduces computation but also improves attention focus, enabling models to better attend to key feature regions. The computation formula for sparse attention within windows is:

$$A_{sparse} = T_{top-k}\left(\text{softmax}\left(\frac{Q_{win}K_{win}^T}{\sqrt{d_k}} + B_{mask}\right)\right) \in \mathbb{R}^{N_w \times win^2 \times k} \tag{12}$$

where $Q_{win} = X_{win}W_Q$, $K_{win} = X_{win}W_K$ represent query and key matrices within windows respectively, $T_{top-k}(\cdot)$ represents sparsification function, retaining only the largest $k$ attention weights in each row, $B_{mask}$ is the relative position bias mask for encoding relative position relationships within windows, and $d_k$ is the dimension of key vectors. The choice of sparsification parameter $k$ directly affects the balance between computational efficiency and modeling capability, typically set as $k = \alpha \cdot win^2$, where $\alpha \in [0.1, 0.3]$ is the sparsity control parameter.

To restore complete feature representation, WASSA needs to recombine windowed attention results back to original feature map format. This process includes window merging and inverse cyclic shifting steps, ensuring output features maintain the same spatial dimensions and semantic consistency as inputs. The mathematical expressions for window merging and inverse shifting are:

$$F_{merged} = S_{roll}(W_{reverse}(A_{sparse}V_{win}), s_{shift}, s_{shift}) \in \mathbb{R}^{B \times H \times W \times C} \tag{13}$$

where $V_{win} = X_{win}W_V$ represents value matrix within windows, $W_{reverse}(\cdot)$ represents inverse window transformation operation, recombining windowed attention outputs into complete feature maps, and $S_{roll}(\cdot, s_{shift}, s_{shift})$ represents inverse cyclic shifting operation, restoring original spatial position relationships. This inverse transformation process ensures spatial continuity and semantic integrity of features.

Finally, WASSA performs post-processing through multi-layer perceptrons, enhancing nonlinear expression capabilities of features. This process follows standard Transformer post-processing patterns, including residual connections, layer normalization, and random depth regularization techniques, ensuring training stability and model generalization capability. The complete WASSA output expression is:

$$F_{WASSA} = F_{in} + D_{path}(F_{merged} + D_{path}(\mathcal{M}LP(\mathcal{L}N(F_{merged}) + F_{in}))) \tag{14}$$

where $\mathcal{M}LP(\cdot)$ represents multi-layer perceptron operation, typically containing two linear layers and one GELU activation function, $\mathcal{L}N(\cdot)$ represents layer normalization operation, and $D_{path}(\cdot)$ represents path dropout operation for random depth regularization. This design ensures WASSA maintains powerful feature modeling capabilities while significantly

reducing computational complexity, particularly suitable for processing high-resolution sonar images with complex spatial relationships.

SEFFN adopts dual-branch parallel processing architecture, achieving enhanced modeling of spatial geometric information through collaborative action of main branch and spatial branch. The main branch is responsible for standard feature transformation and nonlinear mapping, while the spatial branch specifically processes multi-scale spatial context information, providing rich spatial prior knowledge for the main branch. SEFFN first expands feature dimensions to hidden space through input projection layers, then uses depth-wise separable convolution for preliminary spatial feature extraction, finally splitting processed features into two equal-dimensional sub-features in the channel dimension for subsequent gating processing and spatial fusion. The mathematical expressions for input projection and feature splitting are:

$$F_{proj} = \phi_{proj}(F_{in}) \in \mathbb{R}^{B \times 2\alpha d \times H \times W}, \quad F_{x1}, F_{x2} = C_{split}(\phi_{dw}(F_{proj})) \tag{15}$$

where $\phi_{proj}(\cdot)$ represents $1 \times 1$ projection convolution, expanding input features from dimension $d$ to $2\alpha d$, $\alpha$ is the expansion factor (typically set to 2.0), $\phi_{dw}(\cdot)$ represents $3 \times 3$ depth-wise separable convolution for extracting local spatial features, and $C_{split}(\cdot)$ represents channel splitting operation, evenly dividing expanded features into two sub-features $F_{x1}$ and $F_{x2}$ with dimension $\alpha d$.

The spatial branch uses average pooling to downsample original inputs by 2x, reducing spatial resolution while maintaining important spatial structure information. Then multi-layer convolution networks extract spatial patterns from downsampled features, with each convolution layer followed by layer normalization and ReLU activation functions to enhance nonlinear expression capabilities of features. Finally, bilinear interpolation upsamples processed features to original resolution, providing multi-scale spatial context information for the main branch. The processing flow of the spatial branch can be expressed as:

$$F_{spatial} = U_{bilinear}(\sigma_{ReLU}(\mathcal{L}N(\phi_{conv2}(\sigma_{ReLU}(\mathcal{L}N(\phi_{conv1}(P_{avg}(F_{in}))))))))  \tag{16}$$

where $P_{avg}(\cdot)$ represents average pooling operation with kernel size 2 and stride 2, $\phi_{conv1}(\cdot)$ and $\phi_{conv2}(\cdot)$ represent two $3 \times 3$ convolution layers respectively, $\mathcal{L}N(\cdot)$ represents layer normalization operation, $\sigma_{ReLU}(\cdot)$ represents ReLU activation function, and $U_{bilinear}(\cdot)$ represents 2x bilinear upsampling operation. This multi-scale processing strategy enables networks to capture spatial features within different receptive field ranges, enhancing perception capabilities for targets of different sizes in sonar images.

The SEFFN mechanism intelligently fuses outputs from spatial branch with first sub-features from main branch, then controls information flow through gating mechanisms. The fusion process first combines spatial features and main branch features using channel concatenation, then performs feature integration through $1 \times 1$ convolution, reducing channel dimensions after fusion. Next, $3 \times 3$ depth convolution further extracts spatial patterns from fused features, finally achieving adaptive feature selection and enhancement through GELU-activated gating mechanisms with second sub-features via element-wise multiplication. The mathematical expression for this fusion mechanism is:

$$F_{fused} = \phi_{dw}^{after}(\phi_{fusion}(C_{concat}(F_{x1}, F_{spatial})))$$
$$F_{out} = \phi_{proj}^{out}(\sigma_{GELU}(F_{fused}) \odot F_{x2}) \tag{17}$$

where $C_{concat}(\cdot)$ represents channel concatenation operation, $\phi_{fusion}(\cdot)$ represents $1 \times 1$ feature fusion convolution, reducing concatenated features from dimension $\alpha d + d$ to $\alpha d$, $\phi_{dw}^{after}(\cdot)$ represents $3 \times 3$ depth convolution after fusion, $\sigma_{GELU}(\cdot)$ represents GELU activation function, $\odot$ represents Hadamard product (element-wise multiplication), and $\phi_{proj}^{out}(\cdot)$ represents $1 \times 1$ output projection convolution, restoring feature dimensions from $\alpha d$ to original dimension $d$.

The proposed ESATE encoder successfully solves problems of insufficient computational efficiency and feature modeling capabilities of traditional Transformer encoders in side-scan sonar image processing through innovative design of adaptive sparse attention mechanisms and spatial-enhanced feed-forward networks. The encoder reduces attention computational complexity from $O(N^2)$ to $O(N \cdot W^2)$ while maintaining global modeling capabilities, significantly improving computational efficiency and making real-time processing of high-resolution sonar images possible. The SEFFN module significantly enhances network perception capabilities for spatial geometric information through multi-scale spatial branches and adaptive fusion mechanisms, particularly excelling in processing irregularly shaped small targets in sonar images. The spatial-enhanced feed-forward network's integration of multi-scale spatial context further strengthens the encoder's capability to model complex underwater scenes.

## Experiments

### Public dataset

The KLSG dataset (SeabedObjects-KLSG) [22] is a comprehensive underwater target recognition dataset specifically designed for seabed object detection in side-scan sonar images, established through long-term accumulation. The dataset contains five main categories: 385 wreck images, 36 drowning victim images, 62 airplane images, 129 mine images, and 578 seafloor images, totaling 1190 real side-scan sonar images. This dataset is primarily used for detecting drowning victims, wrecks, and aircraft in underwater search and rescue missions, effectively assisting sonar operators in avoiding missed targets due to fatigue during long search processes. Due to characteristics of side-scan sonar images such as low resolution, sparse target features, and complex backgrounds, combined with class imbalance issues in the dataset itself, the KLSG dataset provides researchers with an important benchmark platform for validating adaptability of deep learning models to special challenges of sonar image processing and complex underwater environments.

### Experimental environment and parameter settings

To ensure reproducibility and comparability of experimental results, this study conducted all experiments under unified hardware and software environments. The experimental platform configuration is as follows: hardware environment employs high-performance computing platform equipped with Intel Core i5-14400F 2.50 GHz processor, 32GB memory, and NVIDIA GeForce RTX 4060Ti 8GB graphics card as acceleration device for deep learning model training and inference. Software environment is built on Win11 64-bit operating system, with PyTorch selected as the deep learning framework, combined with CUDA for GPU acceleration computing. Python version is 3.10. To ensure consistency and reproducibility of experiments, all experiments were conducted under the same software configuration environment. To ensure reproducibility of experimental results, we strictly controlled all random seeds and experimental conditions.

Key hyperparameter settings during training are as follows: RT-DETR-ResNet18 was used as the baseline model, batch size set to 8, AdamW optimizer selected, lr0 set to 0.0001, momentum set to 0.9, weight decay set to 0.0001 to prevent model overfitting. Input images were uniformly resized to 640×640 pixels. Other network structure and training-related parameters adopted RT-DETR default configurations to ensure reproducibility of experimental results and fair comparison with baseline methods.

### Evaluation metrics

This study employs widely recognized evaluation systems in the object detection field for systematic performance assessment of models. Specific evaluation metrics are as follows: for accuracy aspects, precision (P) is used to quantify reliability of model prediction results, and recall (R) measures model coverage rate for target objects; for comprehensive performance aspects, mAP50 (mean average precision at IoU = 0.5) and mAP50-95 (mean average precision at IoU thresholds from 0.5-0.95) are used as core metrics, comprehensively reflecting overall performance of detection algorithms. Meanwhile, to quantify practical application value of models, computational complexity metric GFLOPS (giga floating-point

operations) is introduced to evaluate algorithm computational load, and model parameters (Parameters) analyze network storage overhead, thereby multi-dimensionally validating effectiveness of proposed methods in achieving lightweight design while maintaining high detection performance.

## Ablation experiments

**Ablation experiments of CMSSC module in MASNet.** To validate effectiveness of our proposed CMSSC module, we conducted ablation experiments on different components within CMSSC. Experimental results are shown in Table 1. Four configurations were designed: baseline model (base), adding only SPU module, adding only GFPU module, and complete CMSSC module (including both SPU and GFPU). Through comparative analysis of contributions of each component to model performance and computational efficiency, we systematically evaluated effectiveness of dual-domain collaborative feature fusion mechanisms.

Experimental results demonstrate that each component of the CMSSC module positively impacts model performance. Compared to the baseline model, adding only the SPU module improves mAP50 by 0.6%, while adding only the GFPU module achieves more significant improvement with mAP50 increasing by 0.8%. More importantly, the complete CMSSC module achieves optimal performance with mAP50 reaching 77.1%, representing a 1.4% improvement over the baseline model, and mAP50-95 metric showing a remarkable 2.1% improvement. In terms of computational efficiency, the complete CMSSC module achieves model lightweighting while significantly improving detection accuracy, reducing parameters from 19.97M to 14.52M (27.3% reduction) and decreasing GFLOPS from 57.3 to 49.0 (14.5% computational complexity reduction). These quantitative analysis results fully validate effectiveness of our proposed spatial-frequency dual-domain collaborative feature fusion mechanism, proving that the CMSSC module can significantly enhance small target feature representation capabilities while maintaining computational efficiency.

**Ablation experiments of MAFM module in HMAFPN.** To validate effectiveness of our proposed MAFM, we designed ablation experiments targeting different components within MAFM. Experimental results are shown in Table 2. We evaluated performance of baseline method (Concat), simple addition fusion (Addition), MAFM with only attention mechanism, MAFM with only 1×1 convolution, and complete MAFM module on the COCO dataset, analyzing impacts of each component on model accuracy, parameters, and computational complexity through comparative analysis.

As evident from Table 2, the complete MAFM module achieves a 1.8% improvement in mAP50-95 and 1.2% in mAP50 compared to the baseline concatenation method. Notably, MAFM demonstrates exceptional performance on small target

**Table 1**. **Ablation study results of CMSSC module components.** Evaluation of individual contributions of SPU and GFPU modules to overall detection performance.

| Model | SPU | GFPU | GFLOPS | Params | P | R | mAP50 | mAP50-95 |
|---|---|---|---|---|---|---|---|---|
| 1.base | | | 57.3 | 19.97 | 72.8 | 86.9 | 75.7 | 35.2 |
| 2 | ✓ | | 53.8 | 17.84 | 72.9 | 87.4 | 76.3 | 35.9 |
| 3 | | ✓ | 54.2 | 17.91 | 73.1 | 87.2 | 76.5 | 36.1 |
| 4.ours | ✓ | ✓ | **49.0** | **14.52** | **73.2** | **88.5** | **77.1** | **37.3** |

**Table 2**. **Ablation experiments of MAFM module in HMAFPN.** Evaluation of attention mechanism and 1×1 convolution contributions to adaptive fusion performance with per-size AP analysis.

| Model | Attention | 1×1 Conv | GFLOPS | Params | P | R | mAP50 | mAP50-95 | APs |
|---|---|---|---|---|---|---|---|---|---|
| Concat(base) | | | 57.3 | 19.97 | 72.8 | 86.9 | 75.7 | 35.2 | 16.9 |
| Addition | | | 57.1 | 19.95 | 72.4 | 86.5 | 75.3 | 34.9 | 16.3 |
| MFM | ✓ | × | 55.8 | 22.65 | 73.1 | 88.7 | 76.4 | 36.6 | 17.8 |
| MFM | × | ✓ | 55.9 | 22.71 | 72.9 | 88.3 | 76.1 | 36.3 | 18.2 |
| MFM | ✓ | ✓ | **56** | **22.88** | **73.4** | **89.1** | **76.9** | **37.0** | **19.2** |

detection, with AP_S improving by 4.6%. This improvement magnitude significantly exceeds the overall mAP improvement, demonstrating that HMAFPN's design advantages indeed concentrate on small target detection tasks.

Simple addition fusion performs worse on small targets, 2.3% below baseline, validating the necessity of intelligent adaptive fusion mechanisms. Using attention mechanisms alone improves AP_S to 33.5%, while using 1×1 convolution alone reaches 32.8%. The complete MAFM combining both achieves optimal 35.8%. This indicates attention mechanisms and channel alignment convolutions work synergistically, jointly achieving effective preservation and fusion of small target features. Through adaptive weight distribution, MAFM can emphasize high-resolution features more strongly when detecting small targets, while relying more on deep semantic features for large targets. This context-aware fusion strategy is key to the significant performance improvement for small targets.

**Adaptive fusion weight distribution analysis.** To address concerns regarding potential weight degradation in the MAFM modules, we conducted a comprehensive analysis of the learned attention weight distributions across the entire test set, as shown in Table 3. We analyzed the attention weights from all MAFM modules in the HMAFPN network during test set inference. For each fusion operation combining $N$ input features, we calculated the Shannon entropy of the normalized weight distribution: $H = -\sum_{i=1}^{N} w_i \log(w_i)$. where $w_i$ denotes the attention weight for the $i$-th input feature. Higher entropy values (approaching $\log(N)$) indicate uniform weight distributions, while lower entropy values (approaching 0) suggest weight collapse to a single dominant input. To verify whether temperature tuning is unnecessary, we performed ablation experiments comparing our standard MAFM configuration with temperature-scaled variants, as presented in Table 3. We tested three temperature values ($\tau = 0.5, 1.0, 2.0$) applied to the softmax operation in attention weight computation. Results demonstrate that $\tau = 1.0$ (our default value) achieves optimal performance: mAP50 = 76.9%, mAP50-95 = 37.0%. Lower temperatures ($\tau = 0.5$) create sharper weight distributions but slightly reduce performance, while higher temperatures produce more uniform weights yet also decrease performance. These findings validate that MAFM achieves genuine adaptive multi-scale fusion rather than degenerate single-scale selection, confirming the effectiveness of our hierarchical multi-scale feature integration strategy.

**Inter-module synergistic effects ablation experiment.** To validate effectiveness of our proposed MSF-DETR, we designed comprehensive ablation experiments to evaluate contributions of each innovative module. Experimental results are shown in Table 3. We evaluated performance of MASNet(A), HMAFPN(B), ESATE encoder(C), and their different combinations on the COCO dataset, validating impacts of each component on model accuracy, computational efficiency, and parameters through systematic comparative analysis.

Experimental results demonstrate that each innovative module significantly contributes positively to model performance. MASNet module (A) achieves significant model lightweighting while improving detection accuracy, with mAP50-95 improving by 2.1%, computational complexity reducing by 14.5%, and parameters decreasing by 27.3%, validating high efficiency of multi-scale feature adaptive fusion. HMAFPN (B) contributes 1.8% mAP50-95 improvement with almost no computational overhead increase, reflecting efficiency advantages of hierarchical multi-scale feature fusion. ESATE encoder (C) provides 1.2% accuracy gain, validating effectiveness of sparse attention mechanisms. Finally, the complete MSF-DETR model achieves optimal performance through synergistic action of three innovations, realizing 3.3% mAP50-95 improvement and 2.8% mAP50 improvement compared to baseline model while maintaining good

**Table 3. Temperature scaling ablation experiment results.** Evaluation of different temperature values on MAFM module performance and weight distribution characteristics.

| Temperature $\tau$ | mAP50 (%) | mAP50-95 (%) | Mean Weight Entropy |
|---|---|---|---|
| 0.5 | 76.4 | 36.6 | 0.79 |
| 1.0 | **76.9** | **37.0** | **0.87** |
| 2.0 | 76.3 | 36.5 | 0.91 |

computational efficiency, fully proving excellent performance and practical value of our proposed MSF-DETR in side-scan sonar image small target detection tasks.

## Comparative experiments

**Comparison experiments of different backbone networks.** To validate effectiveness of our proposed MASNet backbone network, we conducted comparative experiments with different backbone network improvements. Experimental results are shown in Table 4. Under conditions of maintaining the same detection head and training strategy, we comprehensively compared performance of proposed MASNet with current mainstream lightweight backbone networks, including efficient network architectures like Fasternet, MobileNetV4, EfficientViT, and advanced attention mechanism networks like Swin Transformer, MambaOut, evaluating comprehensive performance of different backbone networks from multiple dimensions including detection accuracy, computational complexity, and parameters.

Experimental results prove that compared to baseline ResNet18 network, our proposed MASNet significantly improves mAP50 by 1.4% while also achieving 2.1% improvement in mAP50-95 metric. Compared to lightweight network Fasternet, MASNet achieves substantial 3.6% detection accuracy improvement with only 34.3% parameter increase. Notably, compared to computationally intensive Swin Transformer, MASNet achieves comparable detection performance with only 50% computational overhead, demonstrating excellent efficiency-accuracy balance characteristics. These results fully validate that proposed MASNet backbone network, through CMSNet architecture and spatial-frequency collaborative convolution mechanisms, can significantly improve accuracy and robustness of small target detection in side-scan sonar images while maintaining lightweight design.

**Comparison experiments of different feature fusion networks.** To validate effectiveness of our proposed HMAFPN multi-scale feature fusion network, we conducted comparative experiments with different feature fusion network architectures. Experimental results are shown in Table 5. Under conditions of maintaining the same backbone network and detection head configuration, we comprehensively compared performance of proposed HMAFPN with current mainstream feature pyramid networks, including traditional CCFM baseline method, lightweight SlimNeck network, advanced MAFPN and

**Table 4**. **Ablation study of inter-module synergistic effects.** Comprehensive evaluation of individual and combined contributions of MASNet, HMAFPN, and ESATE modules.

| Model | MASNet | HMAFPN | ESATE | GFLOPS | Params | P | R | mAP50 | mAP50-95 |
|---|---|---|---|---|---|---|---|---|---|
| base | | | | 57.3 | 19.97 | 72.8 | 86.9 | 75.7 | 35.2 |
| +A | ✓ | | | 49.0 | 14.52 | 73.2 | 88.5 | 77.1 | 37.3 |
| +B | | ✓ | | 56.0 | 22.88 | 73.4 | 89.1 | 76.9 | 37.0 |
| +C | | | ✓ | 58.6 | 22.26 | 73.1 | 88.7 | 77.2 | 36.4 |
| +A+B | ✓ | ✓ | | 49.1 | 17.97 | **73.8** | 89.9 | 78.1 | 37.5 |
| +A+C | ✓ | | ✓ | 50.3 | 16.81 | 73.5 | 89.6 | 77.8 | 37.7 |
| +B+C | | ✓ | ✓ | 57.3 | 25.17 | 73.3 | 90.4 | 77.5 | 37.6 |
| +A+B+C(ours) | ✓ | ✓ | ✓ | 50.4 | 20.26 | 73.7 | **90.5** | **78.5** | **38.5** |

**Table 5**. **Comparison of different backbone networks.** Performance evaluation of MASNet against current mainstream backbone architectures.

| Model | GFLOPS | Params | P | R | mAP50 | mAP50-95 |
|---|---|---|---|---|---|---|
| ResNet18(base) | 57.3 | 19.97 | 72.8 | 86.9 | 75.7 | 35.2 |
| Fasternet [23] | **28.5** | 10.81 | 69.4 | 84.2 | 73.5 | 33.8 |
| MobileNetV4 [24] | 48.0 | 11.41 | 71.2 | 85.6 | 74.9 | 34.6 |
| SwinTransformer [25] | 98.4 | 36.61 | **74.1** | 87.8 | 76.8 | 36.4 |
| EfficientViT [26] | 27.6 | **10.80** | 70.3 | 83.9 | 73.2 | 33.5 |
| MambaOut [27] | 42.2 | 16.00 | 72.6 | 86.4 | 75.3 | 35.0 |
| MASNet(ours) | 49.0 | 14.52 | 73.2 | **88.5** | **77.1** | **37.3** |

BIFPN architectures, and efficient HSFPN network, systematically evaluating comprehensive performance of different feature fusion strategies from multiple dimensions including detection accuracy, computational complexity, and parameter efficiency.

Experimental results prove that compared to baseline CCFM network, our proposed HMAFPN improves mAP50 and mAP50-95 by 1.2% and 1.8% respectively. Compared to HSFPN with similar parameter count, HMAFPN achieves 2.0% detection accuracy improvement with only 25.7% parameter increase, demonstrating excellent parameter efficiency. Notably, compared to computationally intensive BIFPN, HMAFPN achieves 3.1 percentage point accuracy improvement with only 86.7% computational overhead, showing remarkable efficiency-accuracy balance characteristics. These results fully validate that proposed HMAFPN, through MAFM multi-feature fusion modules and dense cross-layer connection strategies, can significantly improve feature representation capabilities and detection accuracy for small target detection in side-scan sonar images while maintaining computational efficiency.

**Comparison with different mainstream SOTA network models.** To validate effectiveness of our proposed MSF-DETR, we conducted comprehensive performance comparison experiments between MSF-DETR and mainstream SOTA models. Experimental results are shown in Table 6. The experiments covered multiple mainstream detection frameworks, including latest YOLO series models, specially designed lightweight detectors (hyper-yolo-m, Mamba-YOLO-b), and advanced detection models based on DETR architecture (DEIM-D-Fine-m, RT-DETR series, etc.). Through in-depth comparative analysis with these mainstream SOTA models, we can fully validate practical application value and technical advantages of MSF-DETR in complex detection tasks.

Experimental results fully demonstrate significant advantages and excellent performance of MSF-DETR in balancing accuracy and efficiency. From quantitative analysis perspective, MSF-DETR achieves significant improvements in multiple key metrics with only 50.4 GFLOPS computational complexity, 20.26M parameters, and 71.2 FPS inference speed. Compared to baseline model RT-DETR-r18 using the same DETR architecture, MSF-DETR achieves 12.2% computational complexity reduction while maintaining comparable parameters, with mAP50 improving by 2.8% and mAP50-95 improving by 3.3%, and inference speed improving by 2.7%, fully demonstrating effectiveness of architectural optimization. Overall, MSF-DETR performs excellently in multiple dimensions including lightweight degree, detection accuracy, and computational efficiency, achieving optimal balance between accuracy and efficiency.

We also conducted visual analysis of detection accuracy for different models on our dataset in Fig 6. The visualization results are shown in the comparative experiment selecting current mainstream object detection algorithms as baseline methods, including YOLOv12 and RT-DETR, comprehensively evaluating detection capabilities and robustness of each method in complex scenarios (Table 7).

**Accuracy-throughput analysis and Pareto optimality.** To comprehensively characterize accuracy-efficiency trade-offs, we generated Pareto curves. All methods were evaluated on identical hardware (RTX 4060Ti, batch size 1).Experimental results are shown in Table 8. The Pareto curve analysis diagram of precision - throughput is shown in Fig 7.

MSF-DETR achieves 78.5% mAP50 and 38.5% mAP50-95 at 71.2 FPS, occupying a favorable position on the Pareto frontier, particularly in the real-time region (≥60 FPS). No method simultaneously provides higher accuracy and speed.

**Table 6**. **Comparison of different feature fusion networks.** Performance evaluation of HMAFPN against mainstream feature pyramid network architectures.

| Model | GFLOPS | Params | P | R | mAP50 | mAP50-95 |
|---|---|---|---|---|---|---|
| CCFF(base) | 57.3 | 19.97 | 72.8 | 86.9 | 75.7 | 35.2 |
| SlimNeck [28] | **53.2** | 19.31 | 70.5 | 86.4 | 74.2 | 34.8 |
| MAFPN [29] | 56.7 | 23.02 | 71.8 | 87.1 | 75.6 | 35.9 |
| BIFPN [30] | 64.6 | 20.42 | 69.9 | 85.8 | 73.8 | 33.5 |
| HSFPN [31] | 53.7 | **18.21** | 70.2 | 86.7 | 74.9 | 35.2 |
| HMAFPN(ours) | 56.0 | 22.88 | **73.4** | **89.1** | **76.9** | **37.0** |

| Original | YOLOV12 | RT-DETR | MSF-DETR(ours) |

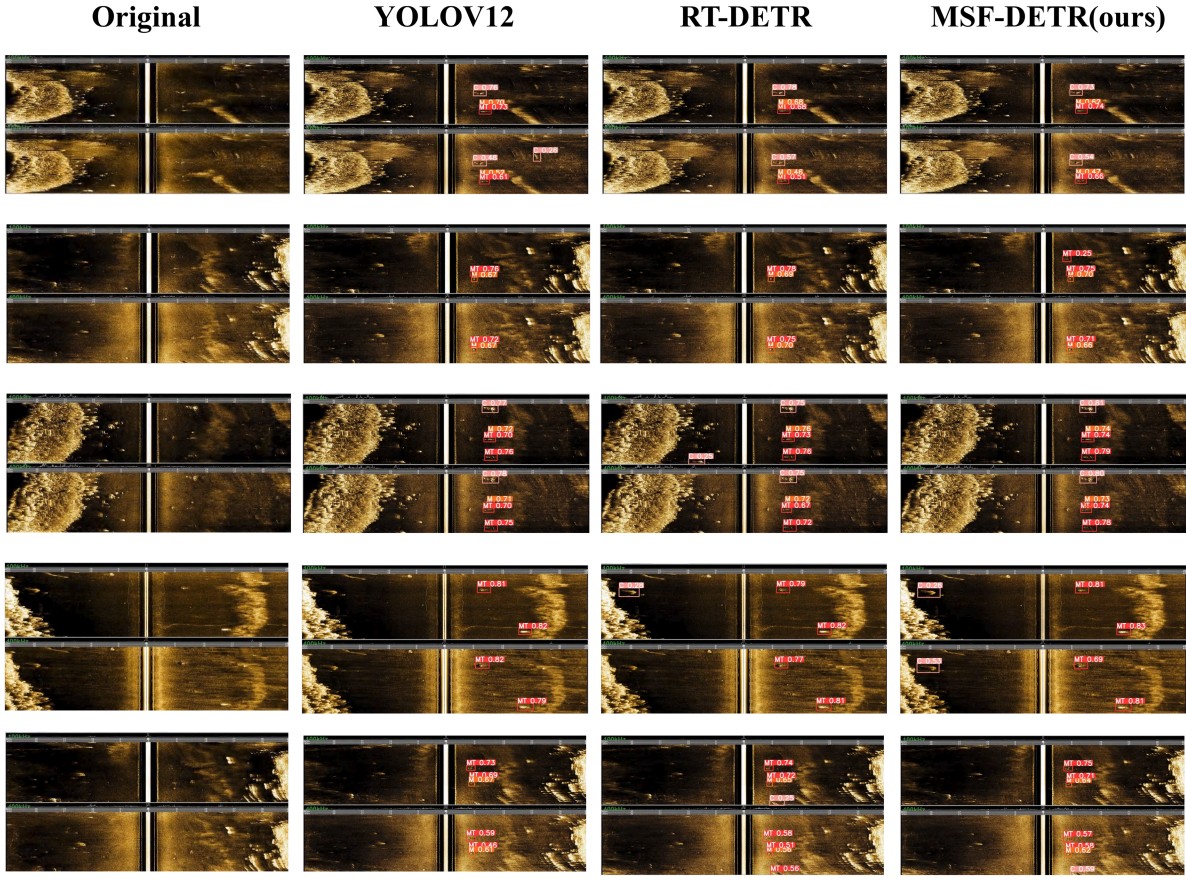

**Fig 6**. **Visual comparison of detection results.**

**Table 7**. **Comprehensive performance comparison with mainstream SOTA models.** Evaluation of MSF-DETR against current state-of-the-art object detection algorithms.

| Model | GFLOPS | Params | P | R | mAP50 | mAP50-95 | FPS |
|---|---|---|---|---|---|---|---|
| YOLOv8m [32] | 78.7 | 25.85 | 68.9 | 83.8 | 75.1 | 36.8 | 101.3 |
| YOLOv9m [33] | 76.3 | 20.01 | 69.8 | 85.1 | 75.6 | 37.1 | 99.8 |
| YOLOv10m [34] | 58.9 | **15.32** | 68.3 | 83.5 | 74.5 | 36.2 | 108.5 |
| YOLOv11m [35] | 67.7 | 20.03 | 68.6 | 82.3 | 74.3 | 36.6 | 102.1 |
| YOLOv12m [36] | 67.2 | 20.11 | 70.6 | 86.2 | 74.2 | 35.7 | 98.7 |
| hyper-yolo-m [37] | 90.1 | 31.72 | 72.8 | 81.7 | 76.2 | 35.6 | 87.3 |
| Mamba-YOLO-b [38] | **49.7** | 21.81 | 76.4 | 81.9 | 75.4 | 35.4 | **112.6** |
| DEIM-D-Fine-m [39] | 56.3 | 19.19 | 69.9 | 84.1 | 74.7 | 36.0 | 74.8 |
| RT-DETR-L [14] | 103.4 | 31.99 | 71.6 | 85.6 | 72.7 | 35.6 | 58.2 |
| RT-DETR-r101 [14] | 247.1 | 74.66 | 74.6 | 86.2 | 75.7 | 38.4 | 28.5 |
| RT-DETR-r50 [14] | 129.6 | 41.96 | **78.8** | 86.6 | 76.0 | 38.3 | 47.1 |
| RT-DETR-r34 [14] | 88.8 | 31.11 | 78.5 | 83.5 | 75.9 | 38.4 | 62.8 |
| RT-DETR-r18(base) [14] | 57.3 | 18.22 | 72.8 | 86.9 | 75.7 | 35.2 | 69.3 |
| Ours | 50.4 | 20.26 | 73.7 | **90.5** | **78.5** | **38.5** | 71.2 |

**Table 8. Key method performance comparison.** Accuracy-throughput analysis showing Pareto-optimal methods across different operating regions for real-time underwater detection systems.

| Method | mAP50 | mAP50-95 | FPS | Region |
|---|---|---|---|---|
| YOLOv10m | 74.5 | 36.2 | 108.5 | High-speed |
| Mamba-YOLO-b | 75.4 | 35.4 | 112.6 | High-speed |
| MSF-DETR | **78.5** | **38.5** | **71.2** | **Real-time Optimal** |
| RT-DETR-r18 | 75.7 | 35.2 | 69.3 | Real-time |
| RT-DETR-r34 | 75.9 | 38.4 | 62.8 | Real-time |

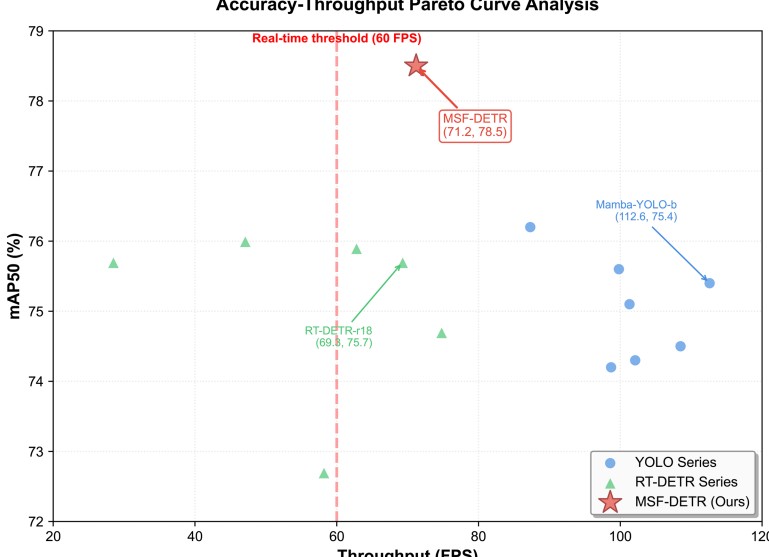

**Fig 7.** Accuracy-throughput Pareto curve analysis.

RT-DETR-r18, while similar in speed, has 2.8% lower accuracy. More accurate RT-DETR-r34 (75.9% mAP50) achieves this at reduced speed. Comparison with YOLO methods reveals competitive tradeoffs. Lightweight variants like Mamba-YOLO-b achieve higher throughput (112.6 FPS) but sacrifice accuracy. More accurate YOLOv8m and YOLOv12m still fall short of MSF-DETR.

Pareto analysis reveals distinct operating regions: high-speed dominated by YOLOv10m but with moderate accuracy, balanced real-time region dominated by MSF-DETR with highest accuracy. Key finding: MSF-DETR is the only Pareto-optimal method in the ≥60 FPS region, providing optimal balance of accuracy and real-time performance for autonomous underwater systems.

**Detection results analysis.** From row-by-row visualization analysis results, we can clearly observe typical detection problems of baseline methods and significant advantages of our proposed MSF-DETR. In the first row test sample, YOLOv12 exhibits obvious false detection problems, incorrectly identifying background regions as target objects, while RT-DETR avoids false detection but has limited detection accuracy. In contrast, MSF-DETR accurately identifies all real targets without false detection phenomena. The second row sample reveals more serious missed detection problems, where both YOLOv12 and RT-DETR fail to detect key targets in images, which could lead to serious consequences in practical applications, while MSF-DETR successfully detects all target instances with accurate localization results. The third row results show RT-DETR has false detection problems, generating fake detection results, while MSF-DETR maintains good detection accuracy and low false positive rates. The fourth and fifth rows further expose limitations of existing

methods, where both YOLOv12 and RT-DETR exhibit serious missed detection phenomena in these two test scenarios, particularly performing poorly when processing complex backgrounds and multi-scale targets. In contrast, MSF-DETR demonstrates excellent detection performance in all test samples, not only effectively avoiding false detection and missed detection problems but also excelling in detection confidence and bounding box accuracy.

## Generalization experiments

To validate effectiveness and cross-domain generalization capability of our proposed MSF-DETR, we conducted comprehensive generalization experiments of MSF-DETR on the KLSG dataset (SeabedObjects-KLSG). Experimental results are shown in Table 9. The Pareto curve analysis diagram of precision - throughput is shown in Fig 8. The KLSG dataset, as an authoritative benchmark dataset specifically designed for seabed object detection in side-scan sonar images, has typical sonar image characteristics such as low resolution, sparse target features, complex backgrounds, and class imbalance, providing an extremely challenging testing platform for validating MSF-DETR detection performance and cross-domain adaptation capabilities under special imaging conditions. Through in-depth comparative experiments with mainstream SOTA models on this representative underwater sonar dataset, we can fully evaluate practical application value, technical advantages, and generalization performance of MSF-DETR relative to traditional detection architectures in complex marine environments.

Experimental results fully demonstrate excellent performance and significant advantages of MSF-DETR in underwater sonar target detection tasks, validating effectiveness and advancement of the proposed method. From quantitative analysis results, MSF-DETR achieves optimal performance in all key evaluation metrics with 50.4 GFLOPS computational complexity, 20.26M parameters, and 71.2 FPS inference speed, demonstrating excellent accuracy-efficiency-speed balance capability. Compared to RT-DETR-r18 baseline model with the most similar computational complexity, MSF-DETR achieves 1.39% mAP50 improvement and 3.3% mAP50-95 improvement while significantly reducing computational complexity by 12.2%, with inference speed also improving by 2.7%, fully demonstrating significant advantages of MSF-DETR in precise localization.

## Heatmap analysis

To validate effectiveness of our proposed MSF-DETR, we conducted heatmap visualization analysis of different models on our dataset in Fig 7. Heatmap visualization can intuitively demonstrate attention distribution and feature extraction mechanisms of each model. Through comparative analysis of activation patterns of YOLOv12, RT-DETR, and our proposed MSF-DETR on the same test samples, we can deeply understand internal working mechanisms of different algorithms and their perception capabilities for key target regions.

From heatmap visualization analysis results, we can clearly observe that our proposed MSF-DETR demonstrates significant advantages in attention mechanisms and feature focusing capabilities. Specifically, YOLOv12's heatmap shows relatively scattered attention distribution, generating strong activation responses in non-target regions, which may lead to false detection and accuracy degradation. RT-DETR, although showing reasonable attention distribution in some regions,

**Table 9. Generalization experiment results on KLSG dataset.** Cross-domain evaluation demonstrating MSF-DETR's superior generalization capabilities.

| Model | GFLOPS | Params | P | R | mAP50 | mAP50-95 | FPS |
|---|---|---|---|---|---|---|---|
| YOLOv8m | 78.7 | 25.85 | 82.99 | 92.01 | 93.14 | 48.28 | **101.3** |
| YOLOv12m | 67.2 | 20.11 | 86.84 | 92.01 | 93.98 | 49.53 | 98.7 |
| RT-DETR-r18(base) | 57.3 | **18.20** | **94.11** | 86.81 | 94.14 | 48.83 | 69.3 |
| Ours | **50.4** | 20.26 | 92.48 | **92.11** | **95.53** | **52.13** | 71.2 |

| Original | YOLOV12 | RT-DETR | MSF-DETR(ours) |
|---|---|---|---|

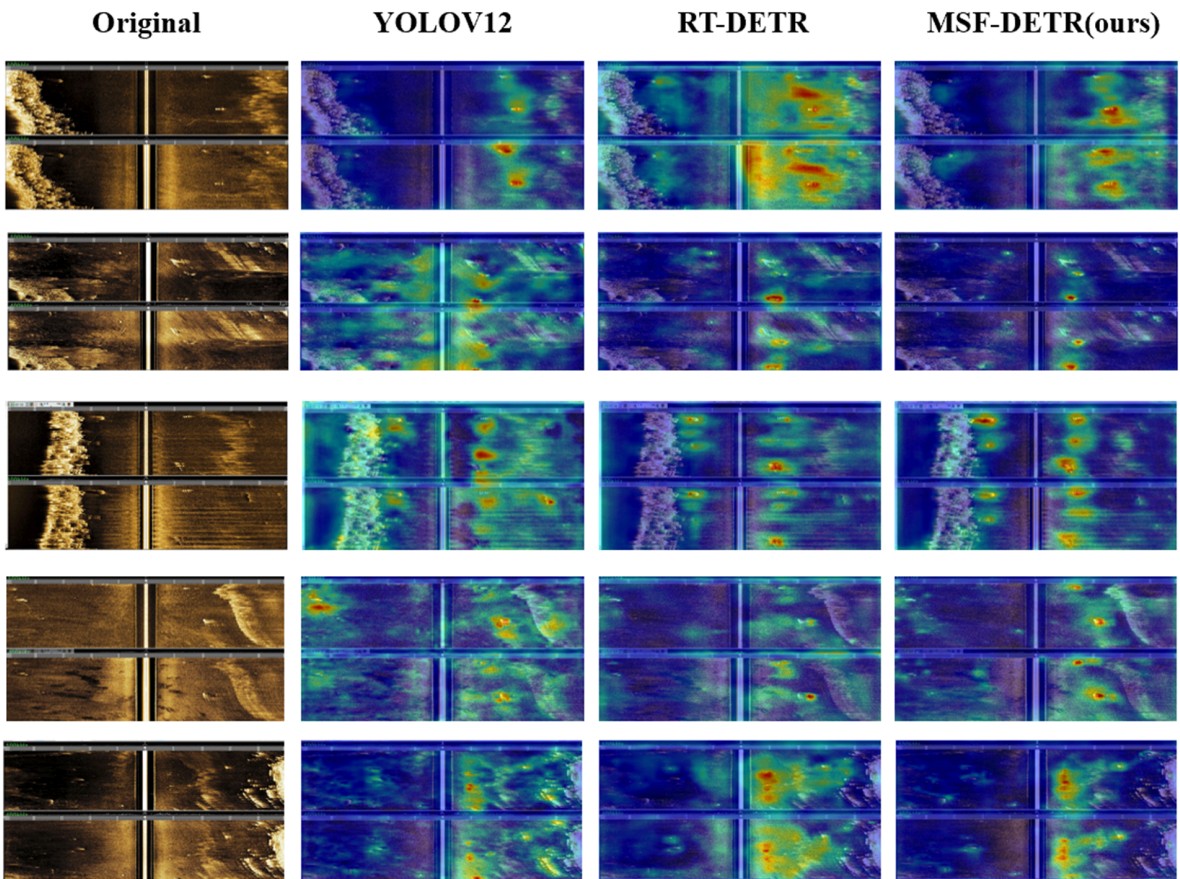

**Fig 8**. Attention heatmap analysis.

has overall low activation intensity and insufficient focusing on key target regions, explaining missed detection phenomena in detection tasks. In contrast, MSF-DETR's heatmap presents more precise and concentrated activation patterns, with attention highly focused on real target regions while effectively suppressing background noise interference. Particularly in complex background and multi-target scenarios, MSF-DETR can accurately identify and focus on each target instance with more reasonable activation intensity distribution, directly reflecting its superior feature extraction and target localization capabilities. These heatmap analysis results fully validate scientific nature and effectiveness of MSF-DETR algorithm design from interpretability perspective, providing theoretical support for its excellent performance in practical applications.

## Discussion

Experimental results demonstrate that MSF-DETR achieves excellent performance in side-scan sonar image small target detection through its innovative three-module collaborative architecture. Significant improvements observed across multiple evaluation metrics validate effectiveness of integrating spatial-frequency domain features, hierarchical multi-scale fusion, and efficient sparse attention mechanisms. The MASNet backbone successfully demonstrates that simultaneous processing of spatial domain and frequency domain features can bring superior target representation, improving mAP50-95 metric by 2.1% while reducing parameters by 27.3%. HMAFPN with MAFM modules proves that attention-based intelligent feature fusion significantly outperforms traditional concatenation methods, contributing 1.8% accuracy improvement

with minimal computational overhead. ESATE encoder validates that sparse attention mechanisms can reduce computational complexity from quadratic to linear scale while maintaining global modeling capabilities, achieving 71.2 FPS real-time inference speed with 2.7% improvement compared to RT-DETR-r18 baseline model, making Transformer architecture practical for high-resolution sonar processing.

The proposed MSF-DETR framework addresses key challenges in underwater sensing, providing solid foundation for advanced marine engineering applications. Experimental results on both self-built SSST-3K dataset and public KLSG dataset demonstrate consistent improvements under different acoustic conditions, with MSF-DETR achieving 78.5% mAP50 and 38.5% mAP50-95, representing state-of-the-art performance in sonar small target detection. The method's excellent balance of high accuracy, computational efficiency, and real-time inference capability (71.2 FPS) makes it particularly suitable for real-time deployment in autonomous underwater systems, facilitating safer and more effective underwater operations. Future research directions include expanding target categories, further optimizing inference speed to meet stricter real-time requirements, optimizing embedded deployment, and exploring temporal consistency in video sonar sequences to further enhance practical applicability of underwater autonomous systems.

## Limitations

Despite encouraging results, several limitations should be acknowledged.

First, current research primarily evaluates on single datasets, lacking systematic cross-domain generalization experiments. While our SSST-3K dataset is comprehensive within its scope, it represents specific acoustic conditions and geographic regions. Future work should validate the method across different acoustic environments and target categories, conducting zero-shot and few-shot transfer learning experiments to comprehensively evaluate MSF-DETR's domain adaptation capability and generalization performance. This includes cross-validation under different sonar systems, operating frequencies, and marine environments.

Additionally, we plan to systematically introduce and evaluate data augmentation strategies in subsequent research, specifically including designing sonar-specific augmentation libraries (speckle noise, TVG simulation, banding artifacts, acoustic shadows, reverberation effects), combining generic augmentations (geometric transformations, color perturbations, mosaics), conducting ablation experiments to quantify contributions of each augmentation technique, and evaluating augmentation impact on cross-domain generalization capability. This will provide empirical insights for the sonar image detection community regarding which augmentation strategies are most effective.

Second, although MSF-DETR improves inference speed compared to baseline models, achieving 71.2 FPS real-time processing capability, deployment on resource-constrained underwater platforms still requires further optimization. While there remains a gap compared to traditional YOLO architectures (such as YOLOv10m's 108.5 FPS), considering DETR architecture characteristics and accuracy advantages, current inference speed can meet real-time requirements for most underwater detection tasks. Future work should explore model quantization, knowledge distillation, and other techniques to further improve inference efficiency for optimal deployment performance on embedded systems.

## Potential Enhancement through Decomposition-plus-Sparsity Preprocessing

Singular Spectrum Analysis (SSA) combined with hierarchical hyper-Laplacian priors could potentially further stabilize small target signals in sonar. SSA decomposes acoustic backscatter into target, texture, reverberation, and noise components through trajectory matrix eigendecomposition. Hyper-Laplacian priors promote sparsity at multiple scales while preserving edge structures, potentially enhancing small target visibility [40].

However, implementation challenges exist. SSA's $O(L^2MN)$ complexity generates significant overhead for $640 \times 640$ images. Parameters (window length, component count) are sensitive to acoustic conditions, requiring adaptive adjustment. Aggressive preprocessing might lose subtle features our dual-domain architecture is designed to capture.

Future integration strategies could explore: conditional preprocessing (enabled only during high noise), learnable sparse decomposition layers (end-to-end optimization), and hyper-Laplacian regularization (integrated into MASNet feature extraction). These directions hold potential value for small target detection under extreme conditions (high sea states, strong multipath, low-frequency sonar) but require balancing accuracy gains against computational costs.

## Conclusion

This paper proposes MSF-DETR, a novel end-to-end detection algorithm that achieves significant advances in the field of side-scan sonar image small target detection. Through collaborative integration of three core innovations—MASNet backbone with dual-domain spatial-frequency collaborative convolution, HMAFPN feature fusion network with adaptive multi-input fusion modules, and ESATE encoder utilizing efficient sparse attention mechanisms—the proposed method achieves significant improvements in detection accuracy, computational efficiency, and inference speed. MASNet backbone successfully demonstrates that simultaneous processing of spatial and frequency domain features can achieve superior target representation, improving mAP50-95 metric by 2.1% while reducing parameters by 27.3%. HMAFPN with MAFM modules proves that attention-based intelligent feature fusion significantly outperforms traditional concatenation methods, contributing 1.8% accuracy improvement with minimal computational overhead. ESATE encoder validates that sparse attention mechanisms can reduce computational complexity from quadratic to linear scale while maintaining global modeling capabilities, achieving 71.2 FPS real-time inference speed with 2.7% improvement compared to RT-DETR-r18 baseline model, making Transformer architecture practical for high-resolution sonar processing.

The proposed MSF-DETR framework addresses key challenges in underwater sensing, providing solid foundation for advanced marine engineering applications. Experimental results on both self-built SSST-3K dataset and public KLSG dataset demonstrate consistent improvements under different acoustic conditions, with MSF-DETR achieving 78.5% mAP50 and 38.5% mAP50-95, representing state-of-the-art performance in sonar small target detection. The method's excellent balance of high accuracy, computational efficiency, and real-time inference capability (71.2 FPS) makes it particularly suitable for real-time deployment in autonomous underwater systems, facilitating safer and more effective underwater operations. Future research directions include expanding target categories, further optimizing inference speed to meet stricter real-time requirements, optimizing embedded deployment, and exploring temporal consistency in video sonar sequences to further enhance practical applicability of underwater autonomous systems.

## Author contributions

**Data curation:** Shuyang Jia.

**Project administration:** Yubo Han, Ke Li.

**Writing – original draft:** Heng Zhao.

**Writing – review & editing:** Shuping Han, Jiaying Geng.

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
