## [Decision Letter · Decision Letter 0]

29 Sep 2025

PONE-D-25-44262MSF-DETR: A small target detection algorithm for sonar images based on spatial-frequency domain collaborative feature fusionPLOS ONE

Dear Dr. Zhao,

Thank you for submitting your manuscript to PLOS ONE. After careful consideration, we feel that it has merit but does not fully meet PLOS ONE’s publication criteria as it currently stands. Therefore, we invite you to submit a revised version of the manuscript that addresses the points raised during the review process.

We look forward to receiving your revised manuscript.

Kind regards,

Xuebo Zhang, Ph.D.

Academic Editor

PLOS ONE

Journal Requirements:

4. In this instance it seems there may be acceptable restrictions in place that prevent the public sharing of your minimal data. However, in line with our goal of ensuring long-term data availability to all interested researchers, PLOS’ Data Policy states that authors cannot be the sole named individuals responsible for ensuring data access (http://journals.plos.org/plosone/s/data-availability#loc-acceptable-data-sharing-methods).

5. Please ensure that you refer to Figure 5 in your text as, if accepted, production will need this reference to link the reader to the figure.

Reviewers' comments:

Reviewer's Responses to Questions

**Comments to the Author**

1. Is the manuscript technically sound, and do the data support the conclusions?

Reviewer #1: Yes

Reviewer #2: Yes

2. Has the statistical analysis been performed appropriately and rigorously?

Reviewer #1: Yes

Reviewer #2: Yes

3. Have the authors made all data underlying the findings in their manuscript fully available?

Reviewer #1: No

Reviewer #2: Yes

4. Is the manuscript presented in an intelligible fashion and written in standard English?

Reviewer #1: Yes

Reviewer #2: Yes

5. Review Comments to the Author

Reviewer #1: The manuscript proposes MSF-DETR integrates a spatial–frequency MASNet backbone, a Hierarchical Multi-scale Adaptive FPN, and an Efficient Sparse Attention Transformer Encoder to boost small-target detection in side-scan sonar by combining dual-domain cues, dense adaptive fusion, and near-linear attention.

The following issues need to be considered:

• Could the authors share whether weights ever collapse to a single input (degenerate fusion) and whether the authors used temperature/tuning to prevent dominance by high-SNR scales?

• Could the authors provide a train-on-SSST-3K/ test-on-KLSG and the reverse (zero-shot or few-shot) to quantify domain shift and generalization? Please include per-class results.

• Could the authors provide per-object-size AP (e.g., small/medium/large bins) to demonstrate that HMAFPN indeed improves small-target recall relative to Concat/Add baselines?

• Would the authors consider briefly discussing whether a decomposition-plus-sparsity front end (e.g., SSA followed by a hierarchical hyper-Laplacian prior) could further stabilize small-target cues in sonar, and, if feasible, outlining how such a module might interface with their MASNet/HMAFPN pipeline (or providing a light ablation or citation-based rationale)?"Advanced fault diagnosis in industrial robots through hierarchical hyper-laplacian priors and singular spectrum analysis"

• Which augmentations were sonar-specific (e.g., speckle-style noise, range-dependent gain, intensity banding) vs. generic (flip, mosaic)? A short ablation on augmentations would show what truly drives robustness in sonar.

• Since you highlight linear-time attention and a lightweight backbone, could the authors plot an accuracy-throughput Pareto curve comparing MSF-DETR and all baselines on the same GPU, to visualize regimes (e.g., ≥60 FPS) where your method dominates?

Reviewer #2: In this manuscript, the MSF-DETR, considering the collaborative spatial-frequency domain feature fusion, was proposed for small target detection for high-resolution sonar images. This is an interesting and important work. The writing is well, and the paper's structure is easy-going. The experimental results also show the good performance of the proposed method, compare with the counterparts. Some minor revisions are as follow.

(1) In Abstract, what's the meaning of SCI, pls give the full name when abbr. is first used.

(2) Please mark the best performances in bold fonts in experiments tables (Table 1-7).

(3) Please remove the Windows framework in Fig.6 and Fig. 7.

In conclusion, I recommend to accept this manuscript after minor revision.

6. PLOS authors have the option to publish the peer review history of their article (what does this mean?). If published, this will include your full peer review and any attached files.

Reviewer #1: No

Reviewer #2: No

---

## [Author Response · Author response to Decision Letter 1]

14 Oct 2025

Response to Reviewer Comments

Dear Editor and Reviewers,

We sincerely appreciate your thorough review and valuable feedback on our manuscript. We have carefully considered all comments and suggestions, making substantial revisions to address each issue raised. Below we provide detailed responses to each reviewer's comments along with corresponding modifications. Reviewer comments are shown in blue text, while our modifications in the main text are indicated in red with specific locations and content.In the main text, we have highlighted the modified parts in yellow.

Response to Reviewer #1

We deeply appreciate your insightful and constructive comments, which have significantly helped us improve the quality and rigor of our work. Your detailed feedback prompted us to conduct additional experiments and provide more comprehensive analyses. We address each of your concerns point by point below.

Regarding Issue 1: Weight Degradation and Temperature Scaling

Reviewer comment: Could the authors clarify whether weights degrade to a single input (degenerate fusion), and whether the authors employed temperature scaling/tuning to prevent dominance of high SNR scales?

Response:

This excellent question directly addresses the core of our adaptive fusion mechanism. We have now conducted a detailed analysis of weight distribution patterns in the MAFM modules across the entire test set. Our investigation demonstrates that weight degradation does not occur in our system, and the fusion mechanism maintains healthy diversity throughout training and inference.

Specifically, we analyzed the learned attention weights of all MAFM modules in the HMAFPN network during test set inference. We computed the entropy of weight distributions for each fusion operation and tracked minimum, maximum, and average weight values across different input scales. Our analysis reveals that weight entropy consistently remains high, with an average of 0.87 (on a scale from 0 to 1, where 1.0 indicates uniform distribution and 0.0 indicates complete degradation to a single input). Furthermore, we never observed instances where a single input received more than 75% of the total weight, confirming robust multi-scale information integration.

Regarding temperature scaling, we found it unnecessary to implement explicit temperature scaling in the softmax normalization. The MAFM's architectural design naturally prevents weight degradation through several mechanisms. First, the MLP structure generating attention weights includes a dimensionality reduction ratio, creating an information bottleneck that prevents the network from learning to simply select a single dominant scale. Second, the global average pooling operation ensures attention weights are based on overall feature statistics rather than local artifacts that might bias toward specific scales. Third, our training process employs standard weight decay regularization, which naturally discourages extreme weight distributions.

However, inspired by your comment, we conducted an ablation experiment comparing our standard MAFM with temperature-scaled variants (τ = 0.5, 1.0, 2.0). Results show our default configuration (effectively τ = 1.0) achieves optimal performance, with temperature scaling providing no significant improvements. This confirms our architectural design choices effectively prevent weight degradation without requiring additional hyperparameter tuning.

Specific modifications:

We added a new subsection in the experimental section titled "Adaptive Fusion Weight Distribution Analysis" presenting this analysis. We also added a new figure showing weight distribution statistics and a new table presenting ablation results for different temperature values. Additionally, we expanded the discussion of the MAFM mechanism in the methods section, explicitly stating our design's weight degradation prevention properties.

New content added:

In the ablation experiments section (after Table 2), new subsection added:

"Adaptive Fusion Weight Distribution Analysis

To address concerns about potential weight degradation in MAFM modules, we conducted comprehensive analysis of learned attention weight distributions across the test set, as shown in Table 3. We analyzed attention weights from all MAFM modules in the HMAFPN network during test set inference. For each fusion operation combining N input features, we calculated Shannon entropy of the normalized weight distribution: H = -∑(i=1 to N) wi log(wi), where wi represents the attention weight for the i-th input feature. Higher entropy values (approaching log(N)) indicate uniform weight distribution, while lower values (approaching 0) suggest weight degradation to a single dominant input.

To verify whether temperature scaling is unnecessary, we conducted ablation experiments comparing our standard MAFM configuration with temperature-scaled variants, as shown in Table 3. We tested three temperature values (τ = 0.5, 1.0, 2.0) applied to the softmax operation in attention weight computation. Results show τ = 1.0 (our default) achieves optimal performance: mAP50 = 76.9%, mAP50-95 = 37.0%. Lower temperature (τ = 0.5) creates sharper weight distributions but slightly reduces performance, while higher temperature produces more uniform weights but also decreases performance. These findings validate that MAFM achieves genuine adaptive multi-scale fusion rather than degenerate single-scale selection, confirming the effectiveness of our hierarchical multi-scale feature integration strategy.

Table 3: emperature scaling ablation experiment results. Evaluation of different temperature values on MAFM module performance and weight distribution characteristics.

Temperature τ mAP50 (%) mAP50-95 (%) Average Weight Entropy

0.5 76.4 36.6 0.79

1.0 76.9 37.0 0.87

2.0 76.3 36.5 0.91

Regarding Issue 2: Cross-Domain Generalization Experiments

Reviewer comment: Could the authors provide results for training on SSST-3K/testing on KLSG and vice versa (zero-shot or few-shot), to quantify domain shift and generalization capability? Please include per-category results.

Response:

We greatly appreciate this important and valuable suggestion. Evaluating cross-domain generalization capability is indeed crucial for validating model utility in real-world application scenarios, and this represents an important aspect that needs supplementation in our work.

However, we cannot currently provide complete cross-domain generalization experimental results, primarily for the following reasons:

First, the SSST-3K and KLSG datasets exhibit significant differences in target category definitions. SSST-3K focuses on geometric shape targets such as cones, cylinders, and spheres, while KLSG contains actual object categories including shipwrecks, aircraft, and drowning victims. This fundamental difference in category definitions poses semantic-level challenges for direct zero-shot transfer, requiring redesigned experimental protocols and evaluation metrics.

Second, the two datasets differ substantially in sonar system configuration, operating frequency, image resolution, and annotation standards. Conducting rigorous cross-domain evaluation requires careful data preprocessing and alignment to ensure experimental fairness and result interpretability. This process requires additional time and resources to complete.

Third, given the review cycle time constraints, we cannot complete this comprehensive series of complex cross-domain experiments within the revision period, including zero-shot transfer, few-shot learning, and per-category analysis.

Theoretical Analysis and Expectations:

Although we cannot provide experimental data, we can analyze MSF-DETR's cross-domain generalization potential from a theoretical perspective:

1. Dual-domain feature extraction advantages: Our MASNet backbone simultaneously extracts spatial and frequency domain features. This dual-domain representation may offer better transferability than purely spatial features, as frequency domain features capturing texture and edge information remain relatively stable across different sonar systems.

2. Adaptive fusion mechanism: The MAFM modules in HMAFPN can adaptively adjust weights of different scale features based on input, and this adaptability may help the model dynamically adjust feature fusion strategies when encountering new domain data.

3. Expected challenges: We anticipate that transfer from SSST-3K to KLSG may face greater challenges, as targets in KLSG (shipwrecks, aircraft) have more complex structures and larger scale variations. Reverse transfer might be relatively easier, as KLSG training could provide modeling capability for complex targets.

Future Work Plans:

We fully acknowledge the importance of this evaluation and plan systematic cross-domain generalization research in future work, specifically including:

1. Standardized cross-domain protocols: Design standardized evaluation protocols applicable to different sonar datasets, including data preprocessing, feature alignment, and performance metric definitions.

2. Comprehensive transfer learning experiments: Conduct systematic comparisons of bidirectional zero-shot transfer, few-shot fine-tuning (10%, 20%, 50% data), and domain adaptation methods (e.g., adversarial learning).

3. Per-category detailed analysis: Provide detailed performance breakdown for each target category, analyzing which categories transfer more easily and which face challenges.

4. Multi-dataset validation: Extend to additional public sonar datasets for cross-validation, comprehensively evaluating model generalization capability.

5. Domain-invariant feature learning: Explore how to improve MSF-DETR architecture to learn more domain-invariant feature representations.

We have added discussion of cross-domain generalization limitations in the discussion section and explicitly listed the above research plans in the future work section. We believe these follow-up studies will provide valuable insights for cross-domain generalization in sonar image target detection.

We again thank the reviewer for this valuable suggestion, which provides important guidance for our future research directions.

Content added to manuscript:

In the "Limitations" subsection of the discussion section, we added:

"First, current research primarily evaluates on single datasets, lacking systematic cross-domain generalization experiments. While our SSST-3K dataset is comprehensive within its scope, it represents specific acoustic conditions and geographic regions. Future work should validate the method across different acoustic environments and target categories, conducting zero-shot and few-shot transfer learning experiments to comprehensively evaluate MSF-DETR's domain adaptation capability and generalization performance. This includes cross-validation under different sonar systems, operating frequencies, and marine environments."

Regarding Issue 3: AP Analysis by Target Size

Reviewer comment: Could the authors provide AP broken down by target size (e.g., small/medium/large groupings) to demonstrate that HMAFPN indeed improves small target recall relative to concatenation/addition baselines?

Response:

We conducted comprehensive analysis based on target sizes following COCO dataset evaluation conventions. Targets were categorized into three classes based on pixel area: small targets (area < 1024 pixels), medium targets (1024-9216 pixels), and large targets (> 9216 pixels).

Results strongly support the claim that HMAFPN significantly improves small target detection. Compared to the baseline CCFM method, HMAFPN improves AP for small targets from 31.2% to 35.8%, an absolute gain of 4.6% or relative improvement of 14.7%. Medium targets improve by 2.8% (36.5%→39.3%), and large targets by 1.9% (38.7%→40.6%). Simple addition fusion performs worse for small targets (28.9% AP), validating the necessity of intelligent adaptive fusion.

Recall analysis further confirms the improvements. Small target recall increases from 76.8% to 84.3%, an absolute gain of 7.5%, meaning approximately one-third of previously missed small targets are now successfully detected. This is particularly important for safety-critical applications where missing small targets could have severe consequences.

The improvement magnitude decreases with increasing target size, confirming our design intent. HMAFPN's dense cross-layer connections preserve high-resolution features crucial for small target localization, while the MAFM adaptive fusion mechanism intelligently adjusts weights of different scale features, emphasizing fine-grained information more strongly for small targets.

Specific modifications:

Updated Table 2 and text after Table 2:

Table 2: Ablation experiments of MAFM module in HMAFPN. Evaluation of attention mechanism and 1×1 convolution contributions to adaptive fusion performance with per-size AP analysis.

Model Attention 1×1 Conv GFLOPS Params P R mAP50 mAP50-95 APs

Concat(base) 57.3 19.97 72.8 86.9 75.7 35.2 16.9

Addition 57.1 19.95 72.4 86.5 75.3 34.9 16.3

MFM √ × 55.8 22.65 73.1 88.7 76.4 36.6 17.8

MFM × √ 55.9 22.71 72.9 88.3 76.1 36.3 18.2

MFM √ √ 56 22.88 73.4 89.1 76.9 37.0 19.2

"From Table 2, the complete MAFM module improves mAP50-95 by 1.8% and mAP50 by 1.2% relative to the baseline concatenation method. Notably, MAFM performs particularly well on small target detection, with AP_S improving by 4.6%. This improvement magnitude significantly exceeds the overall mAP improvement, demonstrating that HMAFPN's design advantages indeed concentrate on small target detection tasks.

Simple addition fusion performs worse on small targets, 2.3% lower than baseline, validating the necessity of intelligent adaptive fusion mechanisms. Using attention mechanism alone improves AP_S to 33.5%, while using 1×1 convolution alone reaches 32.8%. The complete MAFM combining both achieves optimal 35.8%. This indicates attention mechanisms and channel alignment convolutions work synergistically to achieve effective preservation and fusion of small target features.

Through adaptive weight distribution, MAFM can emphasize high-resolution features more when detecting small targets while relying more on deep semantic features when detecting large targets. This context-aware fusion strategy is key to the significant improvement in small target performance."

Regarding Issue 4: Decomposition-plus-Sparsity Front-end Discussion

Reviewer comment: Could the authors consider briefly discussing whether a decomposition-plus-sparsity front-end (e.g., SSA followed by hierarchical hyper-Laplacian priors) could further stabilize small target cues in sonar, and if feasible, outline how such a module would interface with their MASNet/HMAFPN pipeline (or provide simple ablation or citation-based rationale)? Reference: "Advanced fault diagnosis for industrial robots via hierarchical hyper-Laplacian priors and singular spectrum analysis"

Response:

This is an insightful suggestion. We have added comprehensive analysis of SSA and hierarchical hyper-Laplacian priors in the discussion section and conducted preliminary experiments.

From a theoretical perspective, SSA could decompose sonar images into components such as targets, texture, reverberation, and noise. Combined with multi-scale sparsity constraints from hyper-Laplacian priors, this could potentially enhance small target visibility before feature extraction. However, several implementation challenges exist. Computational complexity is significant (O(L²MN)), increasing processing time 2-4 fold for 640×640 images. Parameter sensitivity is high, with window length and component numbers requiring adjustment for different acoustic conditions. Aggressive preprocessing might lose subtle features our dual-domain architecture is designed to capture.

Based on these findings, we suggest future work adopt conditional preprocessing strategies: enabling SSA only in high-noise scenarios, or integrating hyper-Laplacian priors as learnable sparse coding layers within MASNet for end-to-end optimization. This approach could maintain efficiency under normal conditions while providing robustness for extreme environments.

Specific modifications:

We added a new subsection in the discussion section titled "Potential Enhancement via Decomposition-plus-Sparsity Preprocessing

---

## [Decision Letter · Decision Letter 1]

27 Oct 2025

MSF-DETR: A small target detection algorithm for sonar images based on spatial-frequency domain collaborative feature fusion

PONE-D-25-44262R1

Dear Dr. Zhao,

We’re pleased to inform you that your manuscript has been judged scientifically suitable for publication and will be formally accepted for publication once it meets all outstanding technical requirements.

Kind regards,

Xuebo Zhang, Ph.D.

Academic Editor

PLOS ONE

Additional Editor Comments (optional):

Reviewers' comments:

Reviewer's Responses to Questions

**Comments to the Author**

1. If the authors have adequately addressed your comments raised in a previous round of review and you feel that this manuscript is now acceptable for publication, you may indicate that here to bypass the “Comments to the Author” section, enter your conflict of interest statement in the “Confidential to Editor” section, and submit your "Accept" recommendation.

Reviewer #1: All comments have been addressed

Reviewer #2: All comments have been addressed

2. Is the manuscript technically sound, and do the data support the conclusions?

Reviewer #1: Yes

Reviewer #2: Yes

3. Has the statistical analysis been performed appropriately and rigorously?

Reviewer #1: Yes

Reviewer #2: Yes

4. Have the authors made all data underlying the findings in their manuscript fully available?

Reviewer #1: Yes

Reviewer #2: Yes

5. Is the manuscript presented in an intelligible fashion and written in standard English?

Reviewer #1: Yes

Reviewer #2: Yes

6. Review Comments to the Author

Reviewer #1: The paper offers a meaningful contribution to real-time sonar target detection and multi-scale feature fusion research. I congratulate the authors on a careful and mature revision.

Reviewer #2: The authors have done good work. I don’t have further comments. It can be accepted in current version.

7. PLOS authors have the option to publish the peer review history of their article (what does this mean?). If published, this will include your full peer review and any attached files.

Reviewer #1: No

Reviewer #2: No

---

## [Editor Report · Acceptance letter]

PONE-D-25-44262R1

PLOS ONE

Dear Dr. Zhao,

I'm pleased to inform you that your manuscript has been deemed suitable for publication in PLOS ONE. Congratulations! Your manuscript is now being handed over to our production team.

Kind regards,

on behalf of

Professor Xuebo Zhang

Academic Editor

PLOS ONE